

# Wing mechanics and acoustic communication of a new genus of sylvan katydid (Orthoptera: Tettigoniidae: Pseudophyllinae) from the Central Cordillera cloud forest of Colombia

Lewis B. Holmes[1], Charlie Woodrow[1,2], Fabio A. Sarria-S[1], Emine Celiker[3] and Fernando Montealegre-Z[1]

[1] School of Life and Environmental Sciences, University of Lincoln, Lincoln, Lincolnshire, United Kingdom
[2] Department of Ecology and Genetics, Uppsala Universitet, Uppsala, Norbyvägen, Sweden
[3] School of Engineering, University of Leicester, Leicester, United Kingdom

Corresponding author
Fernando Montealegre-Z,
fmontealegrez@lincoln.ac.uk

## ABSTRACT

Stridulation is used by male katydids to produce sound *via* the rubbing together of their specialised forewings, either by sustained or interrupted sweeps of the file producing different tones and call structures. There are many species of Orthoptera that remain undescribed and their acoustic signals are unknown. This study aims to measure and quantify the mechanics of wing vibration, sound production and acoustic properties of the hearing system in a new genus of Pseudophyllinae with taxonomic descriptions of two new species. The calling behaviour and wing mechanics of males were measured using micro-scanning laser Doppler vibrometry, microscopy, and ultrasound sensitive equipment. The resonant properties of the acoustic pinnae of the ears were obtained *via* μ-CT scanning and 3D printed experimentation, and numerical modelling was used to validate the results. Analysis of sound recordings and wing vibrations revealed that the stridulatory areas of the right tegmen exhibit relatively narrow frequency responses and produce narrowband calls between 12 and 20 kHz. As in most Pseudophyllinae, only the right mirror is activated for sound production. The acoustic pinnae of all species were found to provide a broadband increased acoustic gain from ~40–120 kHz by up to 25 dB, peaking at almost 90 kHz which coincides with the echolocation frequency of sympatric bats. The new genus, named *Satizabalus* n. gen., is here derived as a new polytypic genus from the existing genus *Gnathoclita*, based on morphological and acoustic evidence from one described (*S. sodalis* n. comb.) and two new species (*S. jorgevargasi* n. sp. and *S. hauca* n. sp.). Unlike most Tettigoniidae, *Satizabalus* exhibits a particular form of sexual dimorphism whereby the heads and mandibles of the males are greatly enlarged compared to the females. We suggest that *Satizabalus* is related to the genus *Trichotettix*, also found in cloud forests in Colombia, and not to *Gnathoclita*.

## INTRODUCTION

There are over 8,000 species of katydid (Orthoptera: Tettigoniidae) (*Cigliano et al., 2024*) worldwide with many more yet to be described. As one of the few invertebrate groups that communicate acoustically, many of them at ultrasonic frequencies (*Montealegre-Z & Morris, 1999*; *Montealegre-Z, Morris & Mason, 2006*), they are of great interest to researchers. Male katydids primarily rub together their specialised forewings to produce sound by stridulation (*e.g.*, *Ewing, 1989*; *Montealegre-Z & Mason, 2005*) in order to attract a mate or deter a rival. During wing closure, a sharp region on the anal margin of the right forewing (a scraper), engages with a row of teeth (stridulatory file) on the left forewing (*Heller & Hemp, 2014*; *Montealegre-Z, 2009*). The movement of the scraper across the file produces vibrations which are amplified by a large wing cell adjacent to the scraper, the mirror (*Bailey, 1970*). Katydids are known to stridulate with either sustained or interrupted sweeps of the file, generating resonant pure-tone (narrowband frequency) or non-resonant (broadband frequency) calls (*Hemp et al., 2013*; *Montealegre-Z, 2009*). In some katydid species the purity of the call is conserved despite incorporating discrete pulses and silent intervals. This mechanism is exhibited by many Pseudophyllinae, *e.g.*, Cocconotini *Brunner von Wattenwyl, 1895* (*Montealegre-Z & Morris, 1999*; *Montealegre-Z, Morris & Mason, 2006*; *Stumpner et al., 2013*), and is thought that this method of singing evolved as a mechanism to reduce eavesdropping by bats thus reducing the risk of predation (*Heller, 1995*; *Belwood & Morris, 1987*).

The sound reception system in katydids is comprised of paired tympana located in the forelegs. Sound reaches these tympana *via* two pathways (*Bailey, 1990*). The primary route for most species is through the acoustic trachea (*Lewis, 1974*; *Heinrich, Jatho & Kalmring, 1993*; *Michelsen et al., 1994*; *Jonsson et al., 2016*) with sound entering through the acoustic spiracle located in the prothorax (*Nocke, 1975*). The other route to the tympana is external by direct contact with the tympanal surface, and in many katydids this surface is surrounded by auditory pinnae (*Autrum, 1940*, *1942*; *Bailey & Stephen, 1978*; *Mason, Morris & Wall, 1991*). These pinnae have been recently shown to enhance the detection of ultrasonic predators serving as a 'bat detector', by inducing a large sound pressure gain, thus enhancing sound detection of ultrasonic frequencies (*Pulver et al., 2022*; *Woodrow & Montealegre-Z, 2023*). The relic function of the pinnae may have been for protection of the tympanal membrane, which is currently considered to be the ancestral function (*Woodrow, Celiker & Montealegre-Z, 2023*).

Species of the genus *Gnathoclita Haan, 1843*, currently placed in the tribe Eucocconotini *Beier, 1960*, are characterised by having males with large mandibles and a nearly prognathous (forward facing) head orientation. The type species, *G. vorax Stoll, 1813*, is described from Guyana with a distribution throughout Brazil, Suriname, and French Guiana (*Cigliano et al., 2024*). There are two known species described from Peru (*G. izerskyi Gorochov, 2018*; *G. peruviana Carl, 1921*) and a fourth described from South America (*G. laevifrons Beier, 1960*). The fifth and final species, *G. sodalis Brunner von Wattenwyl, 1895*, is described and found throughout Colombia. While the latter species is much smaller in size compared to their congenerics (*Montealegre-Z & Morris, 1999*), no

attempt has been made to resolve the taxonomic status of *G. sodalis*. A related tribe, Cocconotini, holds the genus *Trichotettix Stål, 1873*. The two members, *T. pilosula Stål, 1873* and *T. nuda Beier, 1960*, are characterised by having genicular lobes armed with minute spines and heavily reduced tegmina (*Montealegre-Z & Morris, 1999*). We believe that *G. sodalis* shares more similarities in its morphology and calling song with members of the genus *Trichotettix* than it does with its own congenerics.

This study aims to resolve the taxonomic status of the Colombian species previously assigned to *Gnathoclita*, and to measure and quantify the mechanics of wing vibration and sound production in newly discovered Colombian specimens from the montane forest of the Colombian West and Central Cordilleras. Using these data, we will demonstrate that the Colombian specimens represent a new Cocconotini genus. Using morphological, geographical, and bioacoustic data we compare this new genus with other closely related genera. Additionally, we test the hypothesis that all species here assigned to the new genus *Satizabalus* use a stridulatory mechanism involving wing resonance to produce the observed call and spectral breadth. In other words, the carrier frequency of the call is dictated in part by the natural resonance of the right wing. We also tested the hypothesis that the three species have a similar ultrasonic resonance in the ear pinnae cavities along with acoustic gain for bat detection, comparable to that of other Cocconotini (*e.g.*, *Woodrow & Montealegre-Z, 2023*).

# MATERIALS AND METHODS

## Specimens

A small colony of two of the new species deccribed here, *Satizabalus jorgevargasi* and *S. sodalis*, were kept at the University of Lincoln, UK, in a communal tank in a PHCBI MIR-154 cooled incubator that cycled around a mean temperature of 16.5 °C with a 12:12 h light:dark cycle. They were routinely misted every two days and kept in a humid environment whilst being fed a diet consisting of cut apple, crushed dog biscuit and bee pollen, meal worms, and water, and were kept on a substrate of sphagnum moss with access to dark hides mimicking their natural environment. Here in captivity behavioural ecology, wing mechanics, and bioacoustics could be accurately studied. Two male *S. huaca* specimens were collected in 1997 at La Planada, Colombia; however, they did not survive and could not be recorded. The Colombian Ministry of Environment and Sustainable Development approved permits to conduct fieldwork in Colombian national parks (decree 309 of the 25 February 2000; and DTS0-G-090 14/08/2014). Collected specimens were transported to the University of Lincoln, UK, under collection and exportation permit Expedient PIDB DIG No. 0009-14, auto numero 108 03 Junio de 2014 and No. 00645 09/24/15 (issued by the Colombian Authority of Environment Licences (ANLA)).

The electronic version of this article in Portable Document Format (PDF) will represent a published work according to the International Commission on Zoological Nomenclature (ICZN), and hence the new names contained in the electronic version are effectively published under that Code from the electronic edition alone. This published work and the nomenclatural acts it contains have been registered in ZooBank, the online registration system for the ICZN. The ZooBank LSIDs (Life Science Identifiers) can be resolved, and
the associated information viewed through any standard web browser by appending the LSID to the prefix http://zoobank.org/. The LSID for this publication is: urn:lsid:zoobank.org:pub:0CC2E5D2-6B00-4186-83FD-380790400A78. The online version of this work is archived and available from the following digital repositories: PeerJ, PubMed Central SCIE and CLOCKSS.

## Lab recordings and wing resonance

In the lab, the calls of eight *S. jorgevargasi* n. sp. males were recorded using an ultrasound sensitive 1/8" microphone (frequency range 6.5–140,000 Hz, Brüel & Kjær, Nærum, Denmark) coupled with a Brüel & Kjær 2,633 preamplifier (Brüel & Kjær, Nærum, Denmark), and connected to a G.R.A.S. 12AA 2-channel power module (GRAS sound and vibration, Denmark). Amplifier output was connected to PSV acquisition software (Polytec GmbH, Waldbronn, Germany) which used a National Instruments (NI) acquisition board. A high pass filter was set at 1 kHz, with a sample frequency of 100 kHz. Calls were recorded at a temperature of 19 °C. Males were placed in a wire mesh cylinder suspended 20 cm away from the recording microphone, they were provided with food and water for the duration of the recordings. Acoustic analysis of the calls was done using MATLAB 2022 (MathsWorks, Inc., Natick, MA, USA).

Sound recordings of *S. sodalis* were obtained using a 1/8" Brüel & Kjær Type 4138 condenser microphone (frequency range 6.5–140,000 Hz, cover not removed), connected to a Brüel & Kjær 2,633 preamplifier (Brüel & Kjær, Nærum, Denmark). Data were stored in a notebook computer using an NI USB-6259 board (National Instruments, Austin, TX, USA) and LabVIEW version 9 (32 bit) 2009 software interface (National Instruments, Austin, TX, USA). The microphone's sensitivity was calibrated with a sound level calibrator (Brüel & Kjær, 4,231) and the interface of the Polytec Scanning Vibrometer software (version 8.5; Polytec, Waldbronn, Germany).

Terminology used to describe the temporal patterns of the calling songs have been standardised in accordance with *Baker & Chesmore (2020)*.

Forewing resonance was measured from a single right tegmen from a male of each species (*S. jorgevargasi*, *S. sodalis*, and *S. huaca*), at the University of Lincoln using laser doppler vibrometry (LDV) (PSV-500; Polytec GmbH, Waldbronn, Germany). The tegmen was held in place using a wax made of 50% beeswax (Fisher Scientific, Loughborough, UK) and 50% colophonium (Sigma-Aldrich company Ltd., Dorset, UK). A small amount of the wax was applied to the wing hinge to fix the tegmen in place, the wax was then removed once scanning was complete. Acoustic signals for wing excitation consisted of broadband chirps at 2–60 kHz. These were amplified (A-400; Pioneer, Kawasaki, Japan) and transmitted to a loudspeaker (frequency range 1–120 kHz, Ultrasonic Dynamic Speaker Vifa, Avisoft Bioacoustics, Glienicke, Germany) that was positioned 30 cm in front of the tegmen. Speaker output was corrected to be flat in the frequency range (~1.5–2 dB). The reference signal was recorded using an ultrasound sensitive 1/8" microphone (frequency range 6.5–140,000 Hz, Brüel & Kjær, Nærum, Denmark) positioned next to the tegmen.

## Field recordings

Field recordings of local echolocating bats were taken from Vereda El Chicoral, La Cumbre, Valle del Cauca. An Echo Meter Touch 2 (maximum recording frequency 128 kHz, Wildlife Acoustics, Inc., Maynard, MA, USA) was used along with the Echo Meter App for iOS (Wildlife Acoustics, Inc., Maynard, MA, USA) to record the calls. Echolocation calls were recorded from 19.00 to 20.00, in a transect of approximately 800 m, starting at 3°34′36.7″N 76°35′48.1″W (1799 MASL) and ending at 3°34′18.3″N 76°35′37.4″W (1794 MASL).

## Recording stridulatory wing movements

Stridulatory wing movements and associated sound production were recorded from two male *S. sodalis*. Sound signal was monitored with a B&K 1/4” microphone type 4939 (frequency range 4–100,000 Hz, Brüel & Kjær, Nærum, Denmark) which was connected to a Nexus B&K Conditioning Amplifier (Type 2690; Brüel & Kjær, Nærum, Denmark). Wing movements were recorded using an opto-motion detector (*Hedwig, 2000*). A small piece of reflective tape (Scotchlite 7610 and 8850 retroreflective tape, manufactured by 3M and distributed by Motion Lab Systems Inc., Baton Rouge, LA, USA) was placed on the left forewing and its position monitored with the photodiode of the motion detector. For details of the experiment see *Montealegre-Z & Postles (2010)*.

## Morphological measurements and images

Morphological measurements were taken using digital callipers. The thorax-abdomen length is taken from the most proximal region of the pronotum to the most distal end of the last tergite. Images were taken using an infinite focus system (Alicona G5; Bruker Alicona, Graz, Austria) and a Nikon D7100 DSLR fitted with a Nikon NIKKOR AF-S 50 mm 1:1.8G lens (Nikon Inc., Tokyo, Japan) reverse mounted with a custom-built adapter and external flash (EF-610 DG Super; Sigma Corporation, Kanagawa, Japan). Colour names used in the descriptions of new species were obtained from Werner's nomenclature of colours (*Syme, 1814*).

## Pinnae resonances

Data were collected as previously described in *Pulver et al. (2022)*. Specifically, the ears of *Satizabalus jorgevargasi* n.sp., *S. huaca* n.sp., *S. sodalis* comb. nov, and *Trichotettix pilosula* were µ-CT scanned using a SkyScan 1172 µ-CT scanner (12.9 µm voxel size, 55 kV source voltage, 200 µA source current, 200 ms exposure, 0.2° rotation step, Bruker Corporation, Billerica, MA, USA). For 3D segmentation the scan data were imported into Dragonfly (Comet Technologies Canada Inc., Montréal, Canada) and the pinnae were selected using the circular paint brush every 10 slices followed by interpolation on the Z-axis to connect the slices. The 3D models were then exported as stereolithography (STL) files ready for 3D printing. Printing was done using a Mars Elegoo Pro 2 3D Printer (Elegoo Inc, Shenzhen, China) in a grey photopolymer resin (exposure parameters: 20 s first layer, 5 s normal layers, Elegoo Inc, Shenzhen, China). Once printing was complete, the models were washed in 100% isopropyl alcohol before being left to dry for 24 h.

The 3D printed models of the ears were positioned on a micromanipulator arm with blu-tac (Bostik Ltd, Stafford, UK). A 25mm B&K Type 4182 probe microphone (frequency range 1–20,000 Hz, Brüel & Kjær, Nærum, Denmark), calibrated using a B&K Type 4237 sound pressure calibrator (Brüel &Kjær, Nærum, Denmark), was placed behind the ear, and the ear moved onto the microphone using a micromanipulator, leaving the microphone sticking out of the tympanal cavity 1 cm from the back of the cavity (see video 2 in *Pulver et al., 2022*). In 30 cm away, a loudspeaker (frequency range 500–100,000 Hz, 140-15D Flatfoil, RAAL Advanced Loudspeakers, Serbia), with an amplifier (A-400; Pioneer, Kawasaki, Japan) was positioned in front at the same height as the ear. Stimuli delivered were acoustically scaled to match the wavelength of a real-size ear (*e.g.*, for a model 12x larger than the real ear, the wavelength of the stimulus was 12x larger). This way, the exact frequency delivered to each model would account for variation in printed model scaling. To calibrate the scale of the models, the distance between the pinnae cavities was measured on both the 3D model, and the real ear using Dragonfly (Comet Technologies Canada Inc., Montréal, Canada). Received signals were amplified using a B&K 1708 conditioning amplifier (Brüel &Kjær, Nærum, Denmark), and acquired using a PSV-500 internal data acquisition board (Polytec GmbH, Waldbronn, Germany) at a sampling frequency of 512 kHz. A broadband stimulus of periodic chirps were produced, generated within Polytec 9.4 software (Polytec GmbH, Waldbronn, Germany), with an effective frequency range of 20–200 kHz (accounting for each individual model scale, for example if a model was 12x larger than the original ear then the periodic chirps were downscaled to 1.67–16.67 kHz). Recording in the frequency domain, at a sampling frequency of 512 kHz, the sound pressure level (SPL) of the broadband stimulus was mathematically corrected within the software to deliver 60 dB at all frequencies. From here, the reference frequency spectrum with no ear present could be subtracted from the frequency spectrum reported within the cavities to calculate frequency-specific gain and thus cavity resonance.

## Numerical simulation of cavity resonance

The numerical investigation of the tympanal cavity resonance properties were carried out using the simulation toolbox COMSOL Multiphysics, v. 6.1 (Comsol Multiphysics® v. 6.1, www.comsol.com, Comsol AB, Stockholm, Sweden) and the 3D reconstruction of the μ-CT scanned cavity geometries. The procedures used are outlined in detail in *Pulver et al. (2022)*, hence here we give a brief summary. The scanned geometry of *Satizabalus sodalis* ear cavities was imported into the simulation toolbox COMSOL Multiphysics, v6.1. The boundaries within the cavities representing the tympana were defined manually using the 'form composite faces' function of COMSOL. The tympana were given precise dimensions as measured from the scans of the tympanal membranes: Surface area – 376,733 $\mu m^2$, thickness – 13.09 $\mu m$. The model also incorporated realistic material properties of 3,517 Pa for Young's Modulus, 0.3 for Poisson's ratio and 1,300 kg/m$^3$ for density (*Vincent & Wegst, 2004*; *Celiker, Jonsson & Montealegre-Z, 2020*). The dimensions and material properties incorporated allowed the tympana to have a resonant frequency of

20 kHz. The remainder of the cavity geometry was given a Young's modulus of 2 GPa suitable for insect cuticle (*Vincent & Wegst, 2004*). The cavity geometry was assumed to be located in a free field composed only of air, which was represented by a sphere of radius 3 mm. To avoid reflections from the truncated domain, the boundary condition 'spherical wave radiation' was placed on the boundary of the sphere. The incident wave was modelled to reach the cavities from the front (see *Pulver et al., 2022*). The constructed model was solved in the frequency domain, within the frequency range of 20–200 kHz with a resolution of 0.5 kHz. Details related to the solved mathematical model are given in *Pulver et al. (2022)*. The numerical solution was obtained using the finite element method within COMSOL Multiphysics v6.1.

To test for differences between the cavity resonant frequencies when the model incorporated realistic *S. sodalis* ear properties and when the ear was assumed to have the properties of grey ABS-like photopolymer resin used in 3D printing, we set-up a second numerical model. For the second numerical model, the properties of the tympana were removed and the whole cavity geometry was assumed to have a Young's modulus of 1.8 GPa, which is the actual elastic modulus for ABS-like photopolymer resin. The remainder of the model was kept the same.

## RESULTS

### Taxonomy

*Satizabalus* **n. gen. Holmes and Montealegre-Z**

*Tribe*: Cocconotini *Brunner von Wattenwyl, 1895*

*Etymology*: This genus is named *Satizabalus*, after our colleague, Martin Satizabal, who first discovered the sexual dimorphism within the genus, during a pioneering field trip to the cloud forest of Western Cordillera near Cali, Valle del Cauca, Colombia, in 1996.

*Type species*: *Satizabalus jorgevargasi* n. sp., here described.

*Included species*: *Satizabalus huaca* n. sp., here described., *Satizabalus sodalis* n. comb.

*Diagnosis*: Following the key to tribes in Beier, 1860, *Satizabalus* possess several characters that place this genus within the Cocconotini. The midcoxae lack the elongated ventrobasal tooth (present in Eucocconotini genera). Grooves run mediocaudally from the anterior margin of the mesosternum, male cercus is without a dorsoapical depression, legs are short and compact with the forefemur being 1.5x the length of the pronotum, male subgenital plate with cylindrical styli. In the key to Eucocconotini genera, *Satizabalus* is missing distinctive characters (fully developed wings and the false mirror in female tegmina). In the key to Cocconotini genera, *Satizabalus* advances far and places near *Trichotettix*, *Nastonotus* and *Melanonotus*. Major differences to these genera include enlarged and modified mandibles, vivid green coloration in the legs and tegmina. *Satizabalus* is small in size (25–30 mm body length) whilst *Nastonotus* and *Melanonotus* are much larger (40–60 mm body length). Additionally, the aforementioned genera are commonly found in lowland areas whilst *Satizabalus* is found exclusively in the highland regions of Colombia.

*Description* – Small robust insects. Head large oval shape from the front. Distinct sexual dimorphism present, males have enlarged heads with overlapping asymmetrical mandibles.

Eyes spherical and colour varies between species, primrose yellow in *S. jorgevargasi*, liver brown in *S. sodalis*, greenish black in *S. huaca*. Frontal ocellus circular. Antennae filiform and over three times the length of body. Pronotal edge always ink black. Pronotum granulose. Brightly coloured tegmina and legs. Venation of tegmina bright verdigris green with gambogo yellow perimeter, the same colouration is seen in tibia, fore-, and mid femur. Legs are proportionally short compared to the body, hind femur not extremely long for jumping. Fore tibiae widen and flatten around ear region, paired auditory pinnae are bulbous covering tympana. Both sexes brachypterous. Stridulatory file thick and raised. Male cerci incurved with sclerotised tips, degree of curvature varies between species. Ovipositor stout with no serrations. Calling song audible to humans and resembles a repeated ticking sound.

*Remarks*–There is variation in male head size within species.

### *Satizabalus jorgevargasi* n. sp. Holmes and Montealegre-Z

HOLOTYPE: 1♂ Colombia, Valle del Cauca, Costa Rica, Puente Rojo. 17-XII-14. Collector: F. Sarria-S. Depository: Museo de Entomologia, Universidad del Valle, Cali Colombia. ALLOTYPE: 1♀ Colombia, Valle del Cauca, Costa Rica, Puente Rojo. 5-XI-22 Collectors: C. Woodrow and F. Montealegre-Z. Depository: Museo de Entomologia, Universidad del Valle, Cali Colombia. PARATYPES: 4♂ 3♀ Colombia, Valle del Cauca, Costa Rica, Puente Rojo. 17-XII-14. Collector: F. Sarria-S. 3♂ 2♀ Colombia, Valle del Cauca, Costa Rica, Puente Rojo. 5-XI-22. Collectors: C. Woodrow and F. Montealegre-Z. Depository: University of Lincoln Bioacoustics and Sensory Biology Lab.

*Locality*: Costa Rica: Puente Rojo. This village is located in the municipality of Ginebra, Valle del Cauca (lat. 3°45′05.3″N, long. 76°13′57.8″W).

*Etymology*: This species is named *S. jorgevargasi*, after Jorge Vargas-Sarria who is the owner of the land where this species is most abundant.

*Distribution*: Colombia, Western slope of Central Cordillera cloud forest (Fig. 1).

*Diagnosis*: Eyes are primrose yellow in life, while they are liver brown in *S. sodalis* and greenish black in *S. huaca*. Frons and gena brownish red, Labrum and clypeus pitch black, all much darker in colour than *S. sodalis* whilst lighter than *S. huaca*. Stridulatory file bears 177 teeth, fewer than *S. sodalis*. Front femora, mid-, hind tibiae verdigris green, colour is absent in mid femora and front tibiae where it is present in *S. sodalis*.

*Description*:

*Head* – Wide oval shape in front view, males (Fig. 2A) being much larger than females (Fig. 2B) with highly pronounced overlapping and asymmetrical mandibles. Eyes spherical and primrose yellow. Gena dominates much of head. Mandibles exaggerated. Antenna filiform and over three times the length of body.

*Thorax* – Pronotum granulose, medially compressed and almost as wide as it is long (Fig. 2C). Very short hairs cover pronotum, auditory spiracle circular.

*Legs* – Fore femora begin narrow before widening then tapering distally (Fig. 2H). Fore tibiae with five spines per inner ventral margin. Mid tibiae with five spines per inner

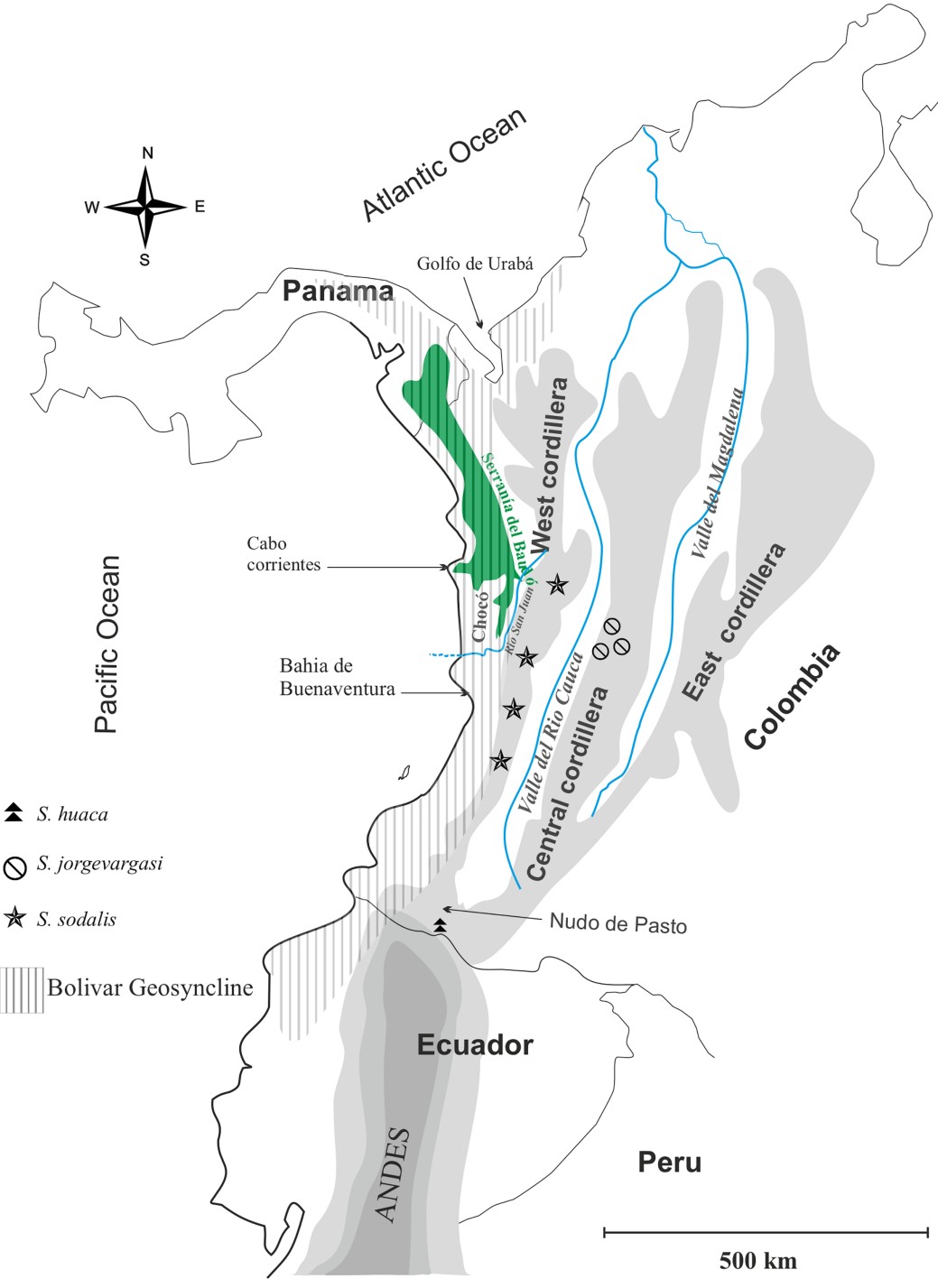

**Figure 1** **Geographical map of Colombia showing the distribution of species in the genus *Satizabalus*.** Image and components created by Fernando Montealegre-Z using CorelDRAW X7 (64-Bit).

ventral margin. Both fore and mid tibiae spines evenly spaced and rear-facing. Hind tibiae with nine spines per inner ventral margin, rear-facing, not evenly spaced. Hind femora with four rear-facing ventral spines, not evenly spaced, increase in size distally (Fig. 2H).

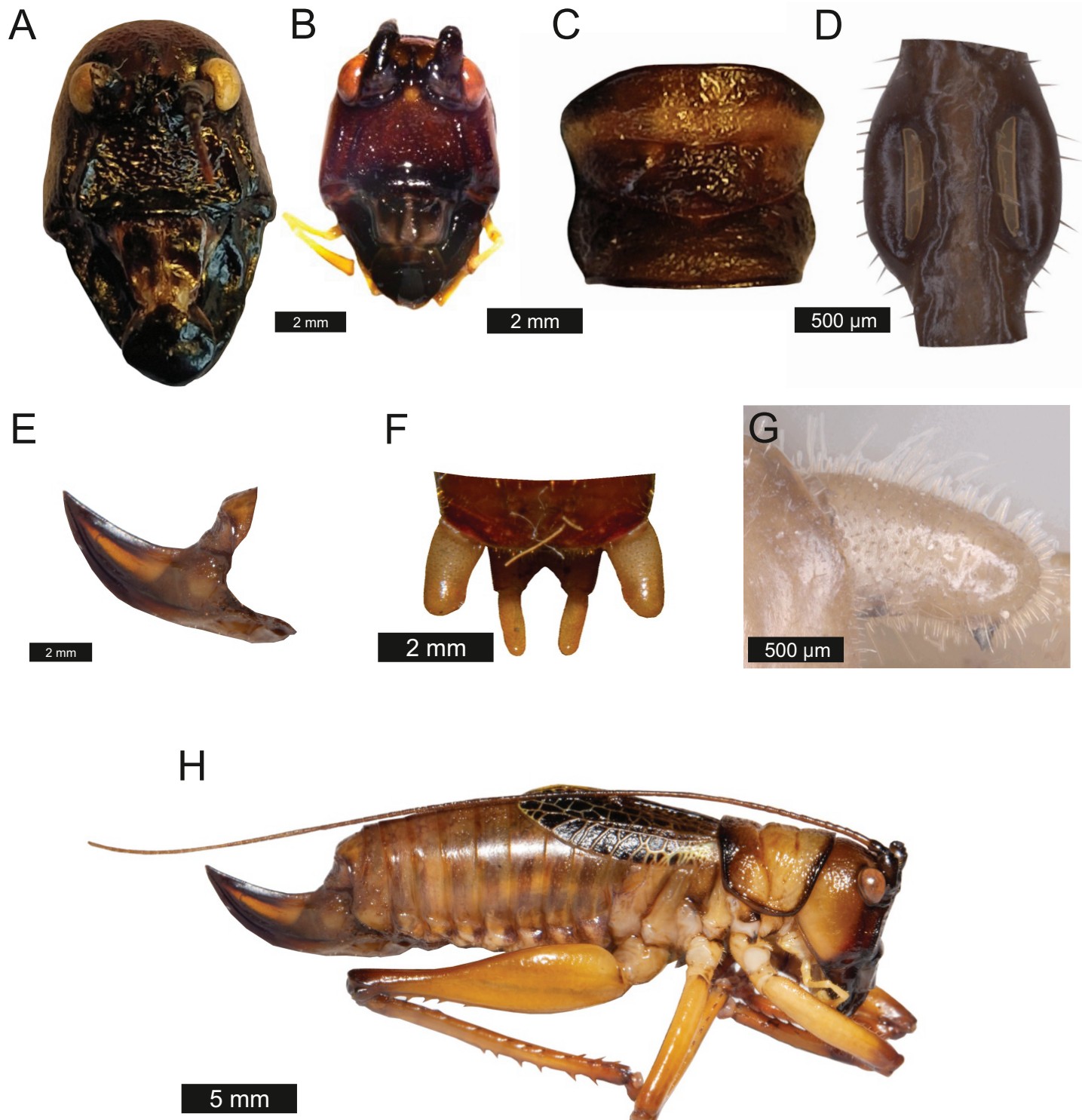

**Figure 2 Morphological characters of *S. jorgevargasi*.** (A) Male head. (B) Female head. (C) Pronotum. (D) Proximal part of right foreleg showing ear region. (E) Ovipositor. (F) Male styli and cerci. (G) Male right cercus. (H) Habitus of female. Photo credit: Charlie Woodrow (F, H) and Lewis B. Holmes (A–E, G).

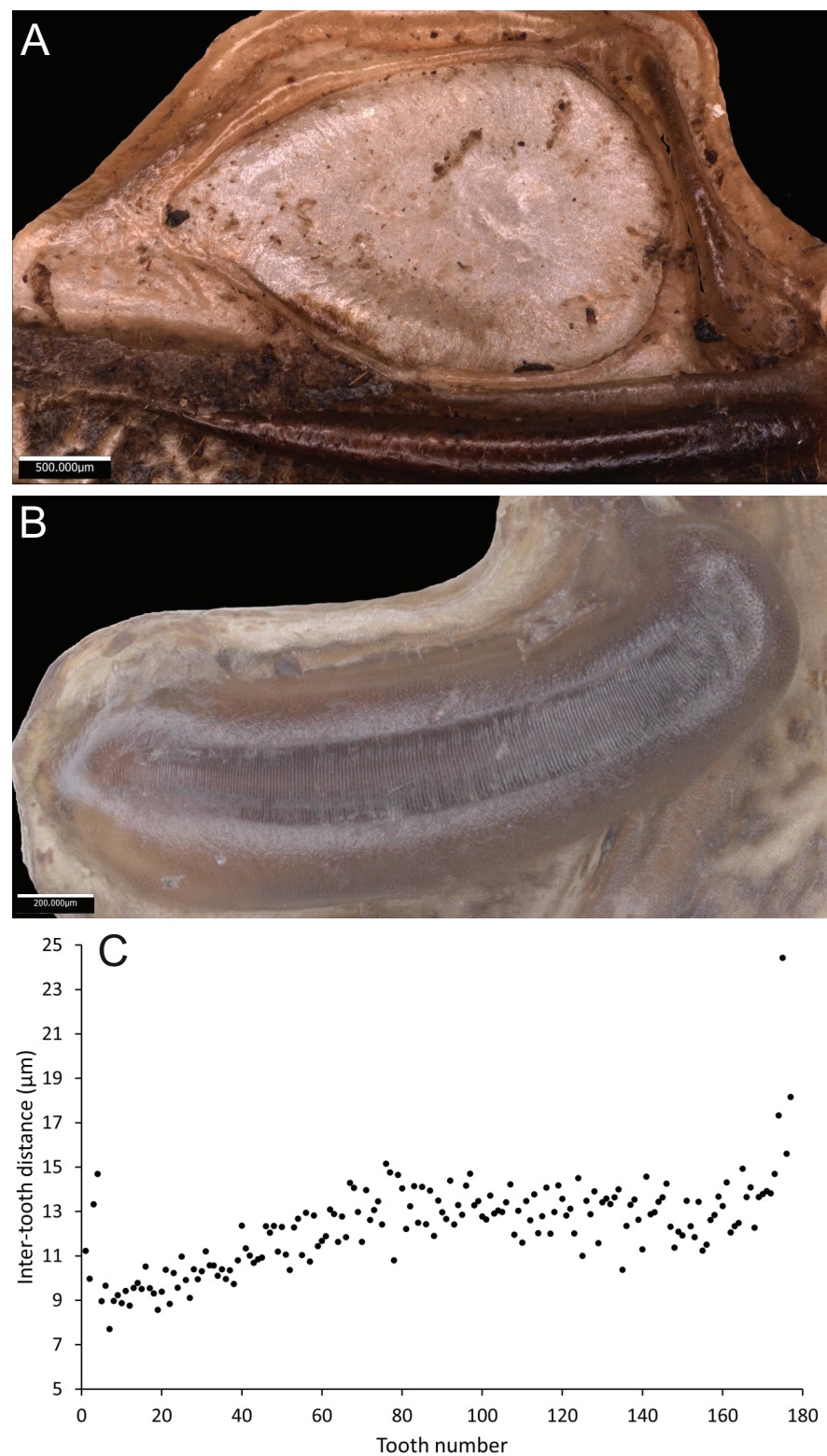

**Figure 3  Stridulatory apparatus of *S. jorgevargasi*.** (A) Mirror of the right tegmen. (B) Stridulatory file displaying tooth distributions. (C) Inter-tooth distances along the length of the file. Images captured by Lewis B. Holmes using an Alicona G5 infinite focus system.

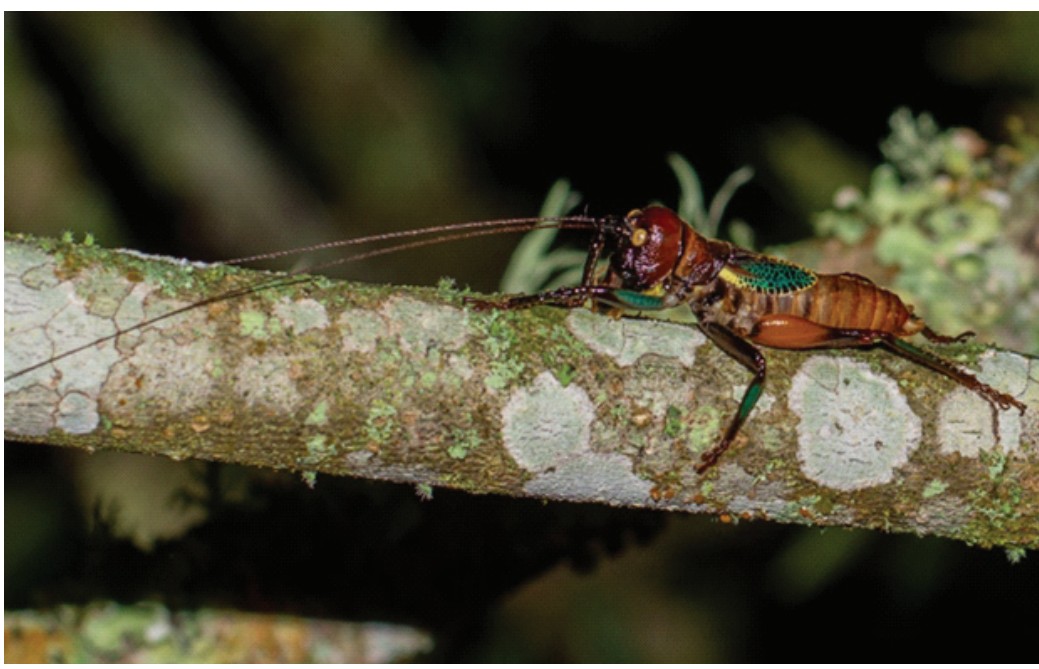

**Figure 4 Habitus of male *Satizabalus jorgevargasi* n.sp. resting on a branch.** Photo credit: Charlie Woodrow, taken *in situ*.                               

*Wings* – Brachypterous. Male tegmen to pronotum ratio 1.8:1 (*n* = 6). Female tegmen to pronotum ratio 1.9:1 (*n* = 6). Tegmen almost twice the length of pronotum. Mirror perimeter of 7.01 mm and area of 3.39 mm$^2$ (Fig. 3A). Stridulatory file on average 2.34 mm long, bears 177 teeth (Fig. 3B).

*Abdomen* – Male cerci marginally incurved, dense hairs and sclerotised tips (Figs. 2F, 2G). Ovipositor approximately third of the length of body (Fig. 2E), curves upwards distally, tapering into fine point.

*Genitalia* – Male subgenital plate with slightly incurved styli. Titillators joined V-shaped, sclerotised and bear very small teeth. Female subgenital plate rounded with large, curved ovipositor approximately a third of the length of body.

*Colouration* – mandibles, scapus, frons, genae, antennal scrobes brownish red. Labrum and clypeus pitch black. In life eyes are primrose yellow. Pronotum buff orange, pronotal edge ink black. Abdomen, hind and mid femora dutch orange. Front femora, mid and hind tibia of the males possess striking verdigris green colouration, absent in females. Tegmina venation verdigris green with edges being gambogo yellow (Fig. 4).

*Measurements* – See Table 1.

*Behavioural ecology* – In the field, *S. jorgevargasi* can been seen on a variety of vegetation and does not exhibit any form of camouflage. Commonly observed on low branches and tree trunks, and around bromeliads. From time to time, females are attracted to the artificial lights of houses. Under laboratory conditions the calling song duty cycle is relatively constant, with males spending most of the night singing from the burrows they guard and protect. They almost always call diurnally, converse to what is observed in the field, which may result from unknown cues in the rearing environment. The exaggerated

**Table 1 Morphological measurements of *Satizabalus* spp.**

| Character | S. jorgevargasi | | | | S. sodalis | | | | S. huaca |
|---|---|---|---|---|---|---|---|---|---|
| | Male (*n* = 6) | | Female (*n* = 6) | | Male (*n* = 5) | | Female (*n* = 5) | | Male |
| | Mean | SD | Mean | SD | Mean | SD | Mean | SD | *N* = 1 |
| Head length | 10.18 | 1.12 | 7.87 | 0.57 | 10.28 | 0.37 | 8.27 | 0.39 | 9.21 |
| Head width | 6.77 | 0.57 | 5.71 | 1.04 | 7.22 | 0.24 | 6.26 | 0.37 | 6.76 |
| Thorax-abdomen length | 15.82 | 2.55 | 18.71 | 2.09 | 14.73 | 0.86 | 19.35 | 1.73 | 14.54 |
| Tegmen | 9.39 | 0.65 | 11.11 | 1.38 | 10.98 | 0.32 | 15.58 | 0.81 | 8.17 |
| Pronotum length | 5.19 | 0.28 | 5.74 | 0.51 | 6.6 | 0.13 | 6.33 | 0.38 | 5.95 |
| Pronotum width | 4.85 | 0.44 | 5.04 | 0.25 | 5.53 | 0.05 | 5.43 | 0.13 | 4.77 |
| Subgenital plate | 3.33 | 0.19 | 2.51 | 0.72 | 3.57 | 0.13 | 2.69 | 0.21 | 2.07 |
| Stridulatory file | 2.34 | 0.09 | N/A | N/A | 2.33 | 0.08 | N/A | N/A | N/A |
| F-femur | 7.28 | 0.61 | 7.58 | 0.48 | 7.65 | 0.33 | 8.39 | 0.48 | 7.84 |
| F-tibia | 7.49 | 0.53 | 7.93 | 0.41 | 7.84 | 0.31 | 8.67 | 0.45 | 8.04 |
| M-femur | 7.05 | 0.46 | 7.54 | 0.31 | 7.82 | 0.19 | 8.51 | 0.36 | 7.77 |
| M-tibia | 7.81 | 0.87 | 8.28 | 0.55 | 8.37 | 0.16 | 9.01 | 0.59 | 8.1 |
| H-femur | 11.25 | 0.65 | 13.02 | 1.21 | 12.48 | 0.47 | 15.61 | 0.43 | 13.13 |
| H-tibia | 11.35 | 0.82 | 13.34 | 0.93 | 12.8 | 0.33 | 15.97 | 0.51 | 13.19 |
| Eye length | 1.71 | 0.07 | 1.77 | 0.18 | 1.86 | 0.09 | 1.84 | 0.06 | 1.81 |
| Eye width | 1.31 | 0.11 | 1.22 | 0.18 | 1.31 | 0.14 | 1.45 | 0.04 | 1.19 |
| Eye depth | 1.44 | 0.11 | 1.44 | 0.15 | 1.53 | 0.12 | 1.66 | 0.15 | 1.41 |
| Cercus | 1.58 | 0.28 | 2.11 | 0.35 | 2.3 | 0.31 | 2.35 | 0.24 | 1.62 |
| Ovipositor | N/A | N/A | 8.19 | 0.69 | N/A | N/A | 9.89 | 0.67 | N/A |
| Stylus | 2.72 | 0.34 | N/A | N/A | 2.34 | 0.11 | N/A | N/A | 2.02 |

**Note:**
All measurements are in mm. F, fore; M, mid; H, hind; SD, standard deviation.

mandibles seen in the males may be an adaption for aggressive behaviours. The purpose of the striking colouring seen in the males is currently unknown, it may be aposematic or possibly a way to convey fitness to a mate. In the lab females were observed depositing eggs into moist sphagnum moss or cotton wool. Eggs take on average 66 days to hatch (*n* = 57) at an average temperature of 16.5 °C, they resemble a grain of rice with an average length of 6.53 mm and width of 1.95 mm (*n* = 10) (Fig. 5A). The nymphs are pistachio green in colouring before darkening to a yellowish brown prior to their next moult (Fig. 5B). They are relatively fast growing, taking roughly 4 months to fully mature.

***Satizabalus sodalis*** (*Brunner von Wattenwyl, 1895*) **n. comb.**

***Gnathoclita sodalis*** *Brunner von Wattenwyl, 1895* **Verh. der Zoologisch-Botanischen Gesellsch. Wien 45:179**
HOLOTYPE: ♂ d492b997-e3df-44a0-894f-eb6fe59acd9a; deposited at: Naturhistorisches Museum, Wien (NMW), Vienna Museum. Locality: Colombia.
*Distribution*: Specimens recorded in this study were found and collected in the Bitaco reserve, located in the Chicoral community in the municipality of La Cumbre, Valle del

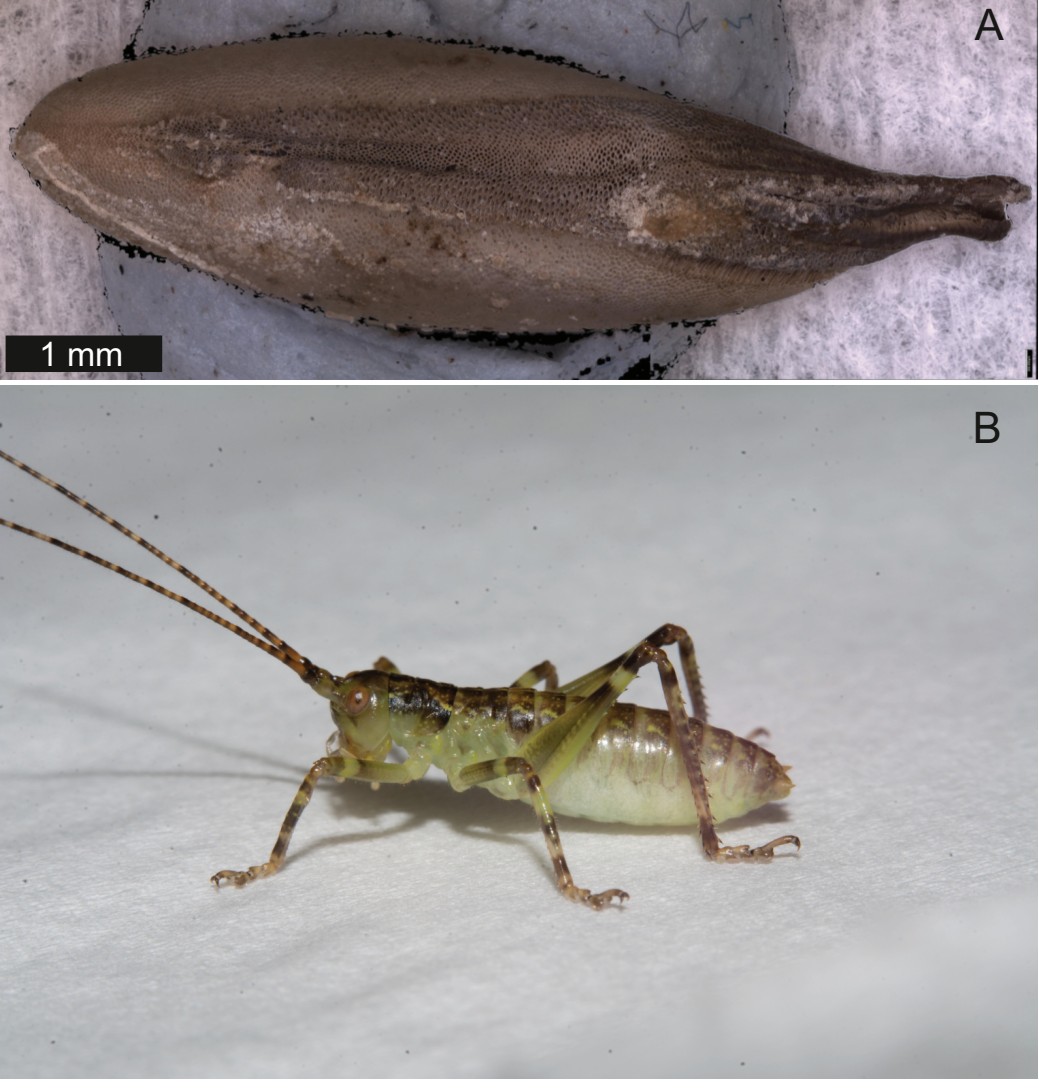

**Figure 5 Life cycle of *Satizabalus jorgevargasi*.** (A) Unhatched egg of *S. jorgevargasi*. (B) Habitus of a newly hatched nymph in the lab. Image credit: Lewis B. Holmes (A) and Charlie Woodrow (B).

Cauca, Colombia (lat. 4° 35′56″N, long. 77° 04″51″W). Other specimens have been found in Colombia: Valle, Dagua, Borrero ayerbe, Bosque El Ensueno, Km. 27 *via* al mar. Elev 1,600–1,800 m. Another specimen from Cauca, Colombia. Orthoptera species file lists their distribution throughout Colombia.

*Diagnosis* – Eyes liver brown in life, compared to primrose yellow in *S. jorgevargasi* and greenish black in *S. huaca*. Genae saffron yellow. Frons orpiment orange. Labrum and clypeus umber brown. Face is lighter in colour than congenerics. Male mandibles rounded where they meet apically, in congenerics they are pointed. Front and mid femora, Front, mid-, hind tibiae verdigris green. Tegmina proportionally longer compared to congenerics however remain brachypterous. Male tegmen to thorax-abdomen length ratio 0.75:1 (*n* = 5) while it is 0.59:1 in *S. jorgevargasi* (*n* = 6) and 0.53:1 in *S. huaca* (*n* = 1). Female

tegmen to thorax-abdomen length ratio 0.81:1 ($n = 5$) while it is 0.59 in *S. jorgevargasi* ($n = 6$). Stridulatory file bears 179 teeth, more than *S. jorgevargasi*.

*Re-Description based on living specimens*:

*Head* – Eyes spherical and liver brown in life. Gena dominates much of head, males possess exaggerated asymmetrical mandibles. Mandibles apically rounded. Antennae filiform and over three times the length of body.

*Thorax* – Pronotum, medially compressed and as wide as it is long (Fig. 6C). Pronotal edge bold. Granulose. Covered in fuzzy hairs.

*Legs* – Fore femora narrow before widening then tapering distally (Fig. 6H). Fore tibiae with five spines per inner ventral margin. Mid tibiae with five spines per inner ventral margin. Both fore and mid tibiae spines evenly spaced and rear-facing. Hind tibiae with nine spines per inner ventral margin, rear-facing, not evenly spaced. Hind femora with four ventral spines, not evenly spaced, increasing in size distally, rear-facing (Fig. 6H). Fore tibiae flatten around ear region (Fig. 6D).

*Wings* – Male tegmen to pronotum ratio 1.66:1 ($n = 5$). Female tegmen to pronotum ratio 2.46:1 ($n = 5$). In females tegmen is over two times the length of the pronotum. Hindwings present but heavily reduced, cannot provide flight. Mirror perimeter 9.02 mm and area 5.42 mm$^2$ (Fig. 7A). Stridulatory file average length 2.33 mm long (Fig. 7B).

*Abdomen* – Male cerci marginally incurved, dense hairs and sclerotised tips (Figs. 6F, 6G). Ovipositor length to width ratio 2.4:1, approximately a third the length of the body (Fig. 6E), curves upwards distally, tapering into fine point.

*Genitalia* – Male subgenital plate dutch orange, slightly incurved styli. Titilators joined U-shape, sclerotised (see Fig. 14 in *Montealegre-Z & Morris, 1999*). Female subgenital plate rounded with large, curved ovipositor approximately a third the length of the body.

*Colouration* – Genae saffron yellow. Scapus, frons, antennal scrobes, and vertex orpiment orange. Mandibles, labrum, and clypeus umber brown. In life eyes liver brown. Pronotum saffron yellow. Pronotal edge ink black. Abdomen, hind, mid femora dutch orange. Front and mid femora, front, mid, hind tibiae in males possess striking verdigris green colouration, absent in females. Tegmina venation verdigris green with edges of tegmina gambogo yellow, remainder of body orpiment orange (Fig. 8).

*Measurements* – See Table 1.

*Behavioural ecology* – In the field, *S. sodalis* is found on a variety of vegetation and does not exhibit any form of camouflage. Commonly observed on low branches and tree trunks, and around bromeliads or at the base of palm leaves. Males spend most of the night singing from the burrows they guard and protect. In the field they are observed to call nocturnally. The exaggerated mandibles seen in the males may be an adaption for aggressive behaviours but may also serve to aid in the excavation of a burrow. The purpose of the striking colouring seen in the males is currently unknown, it may be aposematic or possibly a way to convey fitness to a mate. However, being that the species is nocturnal this is unlikely.

***Satizabalus huaca* n. sp. Holmes and Montealegre-Z**

HOLOTYPE: 1♂ Colombia, Nariño, La Planada. 7-V-97. Collector: F. Vargas. Depository: Museo de Entomologia, Universidad del Valle, Cali Colombia.

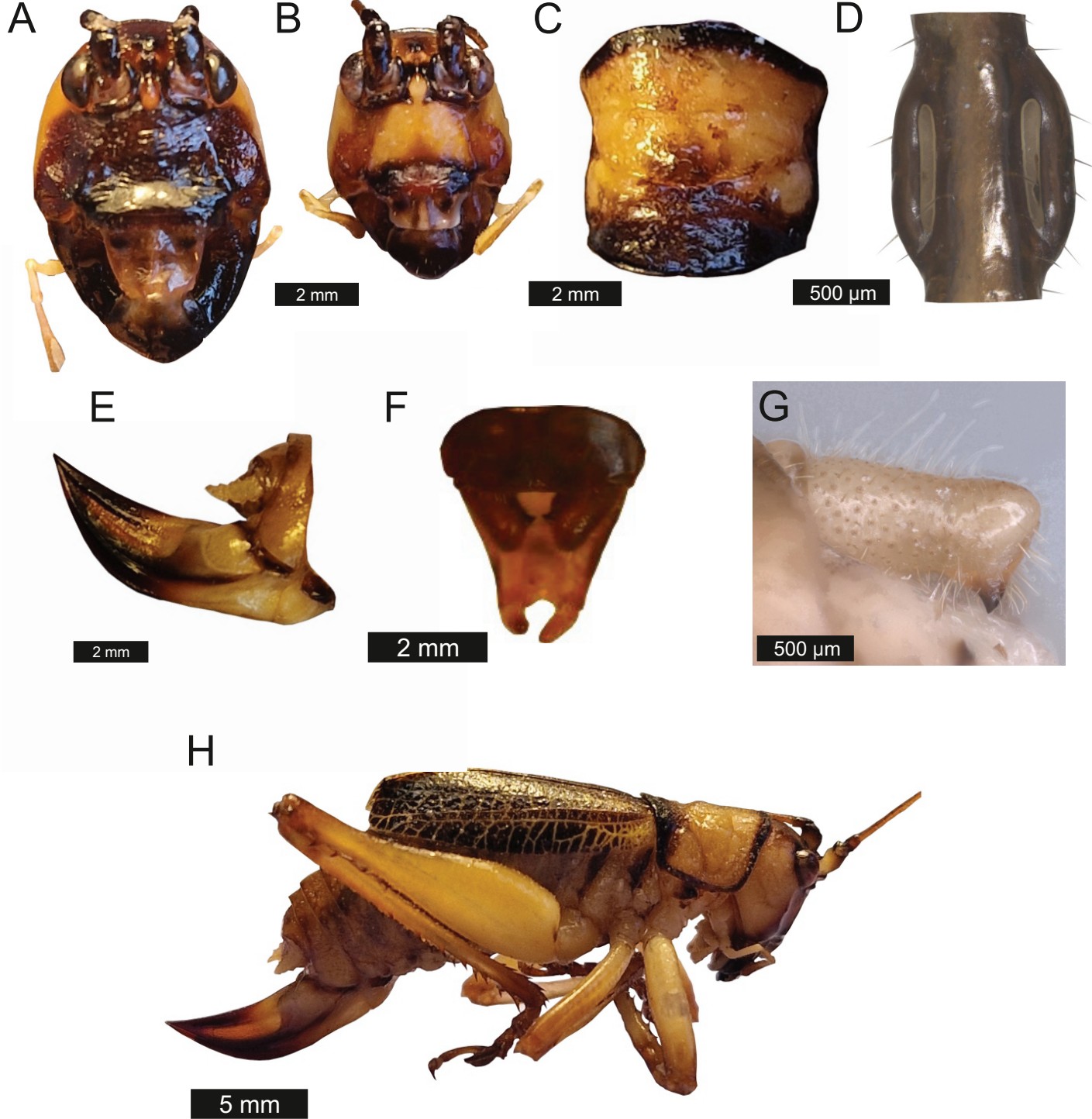

**Figure 6 Morphological characters of *S. sodalis*.** (A) Male head. (B) Female head. (C) Pronotum. (D) Proximal part of right foreleg showing ear region. (E) Ovipositor. (F) Male styli and cerci. (G) Male right cercus. (H) Habitus of female. Photo credit to Lewis B. Holmes.

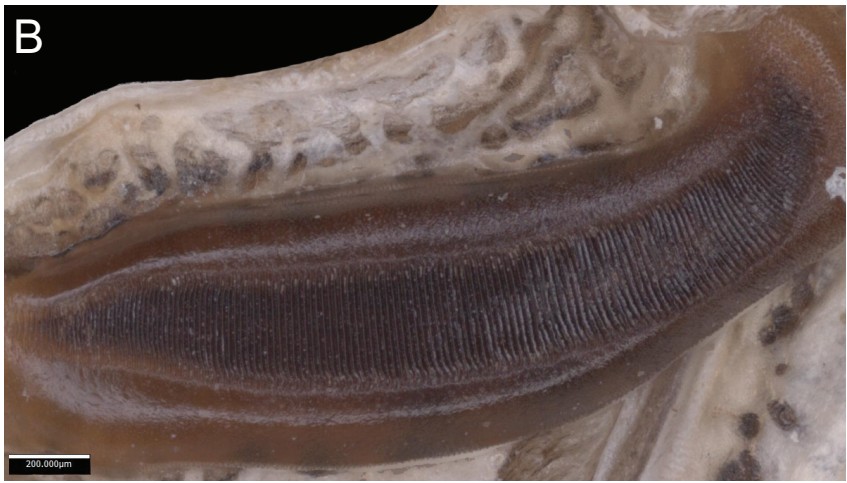

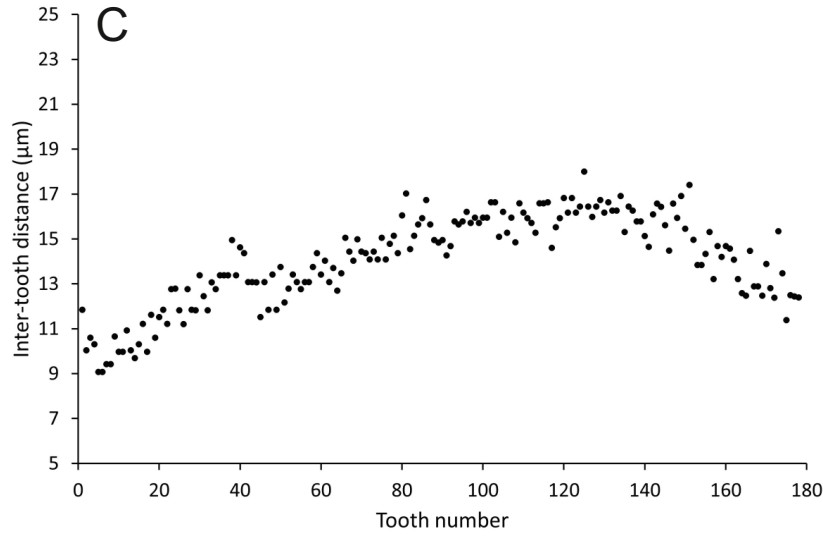

**Figure 7 Stridulatory apparatus of *S. sodalis*.** (A) Mirror of the right tegmen. (B) Stridulatory file displaying tooth distributions. (C) Inter-tooth distances along the length of the file. Images captured by Lewis B. Holmes using an Alicona G5 infinite focus system.

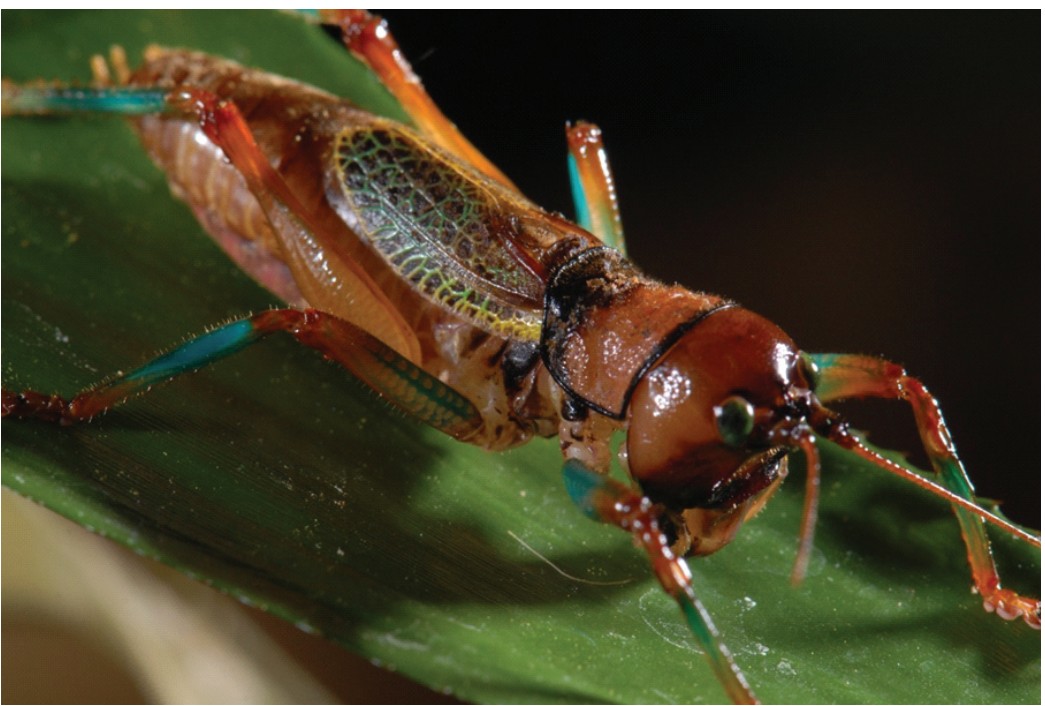

**Figure 8 Habitus of a male *Satizabalus sodalis* resting on a leaf.** Photo credit: Fernando Montealegre-Z taken *in situ*.

*Etymology*: The species name '*huaca*' refers to the location at which it is found and means 'sacred place' in the Incan tradition. The site at which our specimen was found, La Planada (Colombia), is located at the Nudo de los Pastos o Macizo de Huaca, which was a sacred region during the time of the Incas.

*Distribution*: Colombia, montane cloud forest (Fig. 1).

*Diagnosis* – Smaller overall size (23 cm body length) compared to *S. jorgevargasi* (25–30 cm body length) and *S. sodalis* (27–30 cm body length). In life eyes greenish black, while they are primrose yellow in *S. jorgevargasi* and liver brown in *S. sodalis*. Frons, labrum, clypeus reddish black. Genae brownish red. Face much darker in colour compared to congenerics. Front femora and mid tibiae verdigris green, colour absent from mid femora and front and hind tibiae. Mid tibiae only have four spines per inner ventral margin, congenerics have five. Tegmina heavily reduced and brachypterous. Male tegmen to thorax-abdomen length ratio 0.53:1 ($n = 1$) while it is 0.59:1 in *S. jorgevargasi* ($n = 6$) and 0.75:1 in *S. sodalis* ($n = 5$).

*Description*:

*Head* – Wide oval shape from frontal view. Eyes spherical and greenish black in life. Gena account for much of head, males possess exaggerated asymmetrical mandibles. Antennae filiform.

*Thorax* – Pronotum medially compressed and as wide as is long (Fig. 9B) Bold pronotum edge covered in fuzzy hairs.

*Legs* – Fore femora narrow before widening then tapering distally (Fig. 9E). Fore tibiae with five spines per inner ventral margin. Mid tibiae with four spines per inner ventral

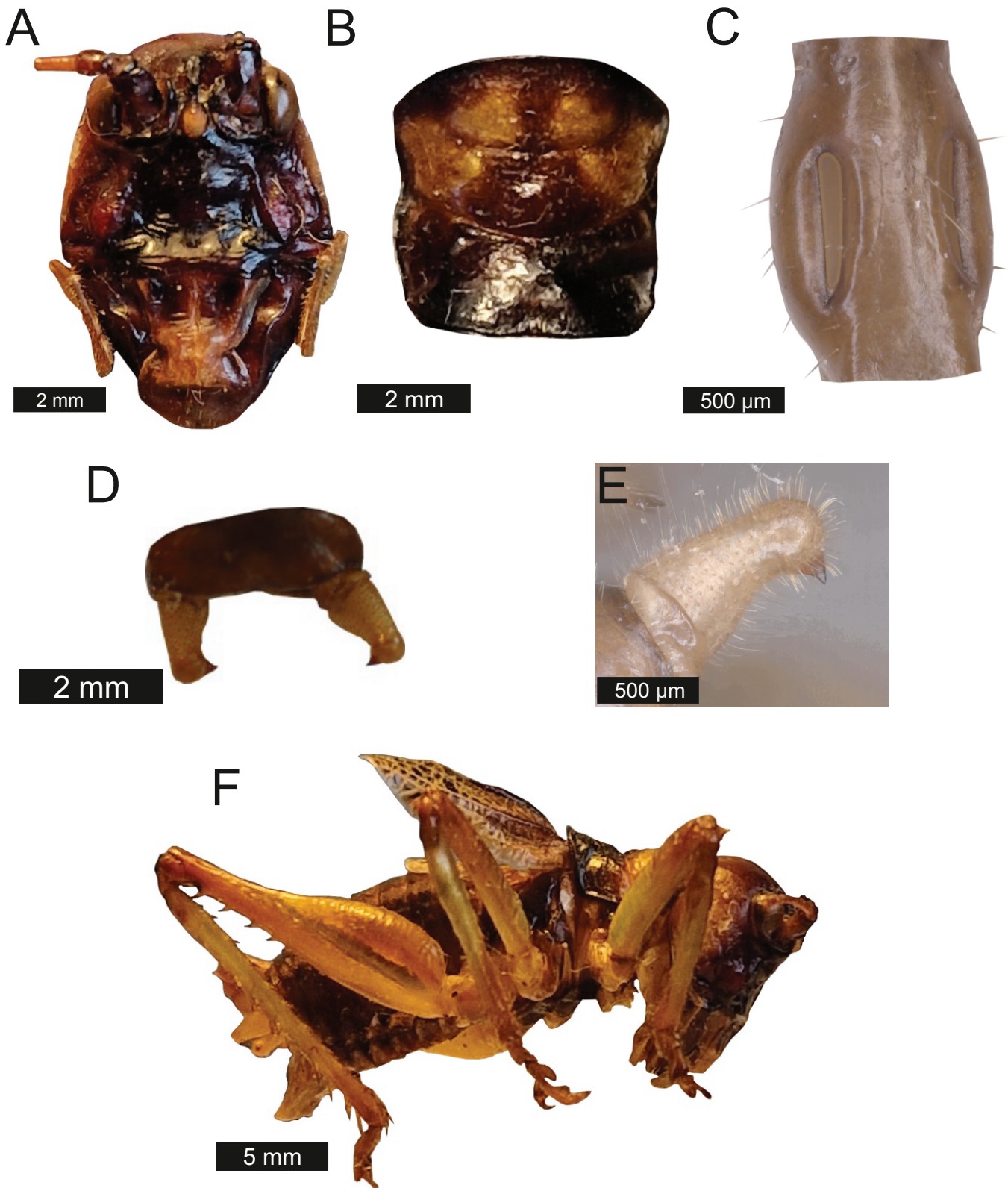

**Figure 9 Morphological characters of *S. huaca*.** (A) Male head. (B) Pronotum. (C) Proximal part of right foreleg showing ear region. (D) Male cerci. (E) Male right cercus. (F) Habitus of male. Photo credit: Lewis B. Holmes.

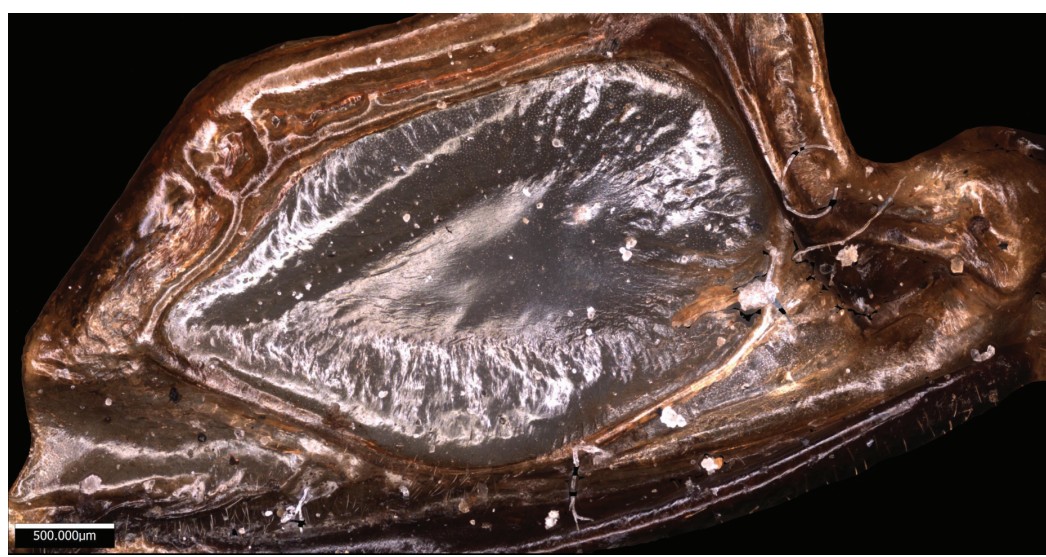

**Figure 10 Stridulatory apparatus of _S. huaca_.** Mirror of the right tegmen. Image captured by Lewis B. Holmes using an Alicona G5 infinite focus system.

margin. Both fore and mid tibiae spines evenly spaced and rear-facing. Hind femora with four ventral spines rear-facing, not evenly spaced, increasing in size distally. Hind tibiae with nine spines rear-facing, not evenly spaced.

_Wings_ – Brachypterous. Male tegmen to pronotum ratio 1.37:1 ($n$ = 1), tegmen are about half the length of the abdomen. Hindwings present but heavily reduced, cannot provide flight. Mirror perimeter 8.04 mm and area 4.4 mm$^2$ (Fig. 10).

_Abdomen_ – Male cerci marginally incurved with dense hairs and sclerotized tips (Fig. 9D).

_Genitalia_ – Male subgenital plate with slightly incurved styli.

_Colouration_ – Frons, antennal scrobes, scapus, and vertex reddish black. Genea, mandibles, labrum, and clypeus brownish red. In life, eyes greenish black. Pronotum deep orange-coloured brown. Pronotal edge ink black. Abdomen liver brown. Front femora, mid-, hind tibia of males verdigris green colouration, absent in females. Tegmina venation verdigris green with edges of tegmina gambogo yellow, remainder of body dutch orange.

_Measurements_ – See Table 1.

## Key to species of _Satizabalus_

1. Labial palps pale, nearly white. Eyes primrose yellow in life. Valle del Cauca Central Cordillera distribution..………………………………………….….. _S. jorgevargasi_

1'. Labial palps and eyes different in colour. Distributed in Western Cordillera or Nudo de Pasto……………………………………………………………………… 2.

2. Smaller overall size compared to congenerics, thorax-abdomen length 14.5 mm in _S. huaca_, 15.8–18.7 mm in _S. jorgevargasi_, 14.7–19.4 mm in _S. sodalis_. In life eyes greenish black, labial palps liver brown. Tegmina heavily reduced and brachypterous. Male tegmen to thorax-abdomen length ratio 0.53:1 ($n$ = 1). Front femora and mid tibiae verdigris green. …………………………………….…………………………………..………………_S. huaca_

2'. Bulky body in males and females. In life eyes liver brown, labial palps umber saffron yellow. Tegmina proportionally longer compared to congenerics however remain brachypterous. Male tegmen to thorax-abdomen length ratio 0.75:1 ($n = 5$). Female tegmen to thorax-abdomen length ratio 0.81:1 ($n = 5$). Front and mid femora, Front, mid-, hind tibiae verdigris green …………………………………..………………. *S. sodalis*

## Biology in captivity

*Satizabalus jorgevargasi* thrives best when kept in cooler temperatures around an average of 16.5 °C with a 12-h day-night cycle. A diet of crushed dog biscuit and bee pollen, cut apple or carrot, water, and meal worms is ideal. Females will typically lay eggs in damp sphagnum moss or moist cotton wool. Eggs take an average of 66 ($n = 57$) days to hatch at a constant temperature of 20 °C. Nymphs require around four months ($n = 18$) to fully mature if fed a high protein diet. Males and females were kept in a communal tank containing a maximum of four males and four females. Adults will cannibalise dead conspecifics. Males will sing almost exclusively at night with the exception of some brief calls being produced when the tanks are misted with a water spray bottle. Nymphs will employ crop fluid regurgitation as a defence upon physical manipulation.

## Acoustic signals

Analysis of the calling song of *Satizabalus jorgevargasi*, recorded in the lab at 19 °C, revealed that this species produced a repeated rapid-decay song structure. The average duration of a song (a single syllable repetitive to the human ear) was 37.4 ± 7 ms (Figs. 11A, 11B). Each syllable contained 9–10 short discrete pulses with a sinusoidal nature tendency (Fig. 11B). Between each syllable there is a silent interval of 97 ± 6 ms. An average of 276 syllables are produced per minute at a carrier frequency of 18.7 ± 1.4 kHz. At −30 dB, spectral breadth ranged from 15–24 kHz. The maximum amplitude of the song was found to be 103 dB (at 5 cm re 20 µPa).

Analysis of the calling song of *S. sodalis*, recorded in the lab at 19 °C, show that this species produced a similar repeated rapid-decay song structure. The average duration of a song was 42.3 ± 4.7 ms (Figs. 12A, 12B). Each syllable contained 18–20 short discrete pulses with a sinusoidal nature tendency (Fig. 12B). Between each syllable there is a silent interval of 109 ± 8 ms. An average of 226 syllables are produced per minute at a carrier frequency of 14.3 ± 1.2 kHz. At −30 dB, spectral breadth ranged from 13–19 kHz. Tracking of the wing motion of *S. sodalis* reveals that sound is only produced during wing closure (Fig. 13). The maximum amplitude of the song was found to be 94 dB (at 10 cm re 20 µPa).

## Wing resonance

Analysis of sound recordings and wing vibrations for *S. jorgevargasi* and *S. sodalis* revealed a clear relationship between mirror vibration and song carrier frequency. *S. jorgevargasi* has a peak wing resonance at 21.8 kHz (Fig. 14D) with a peak song carrier frequency of 18.7 kHz, whilst *S. sodalis* has a peak of 16.2 kHz (Fig. 15D) with a peak song carrier frequency of 14.3 kHz, 2–3 vibrational modes were observed with the most intense mode dictating the calling song frequency. This relationship allows us to predict the song carrier

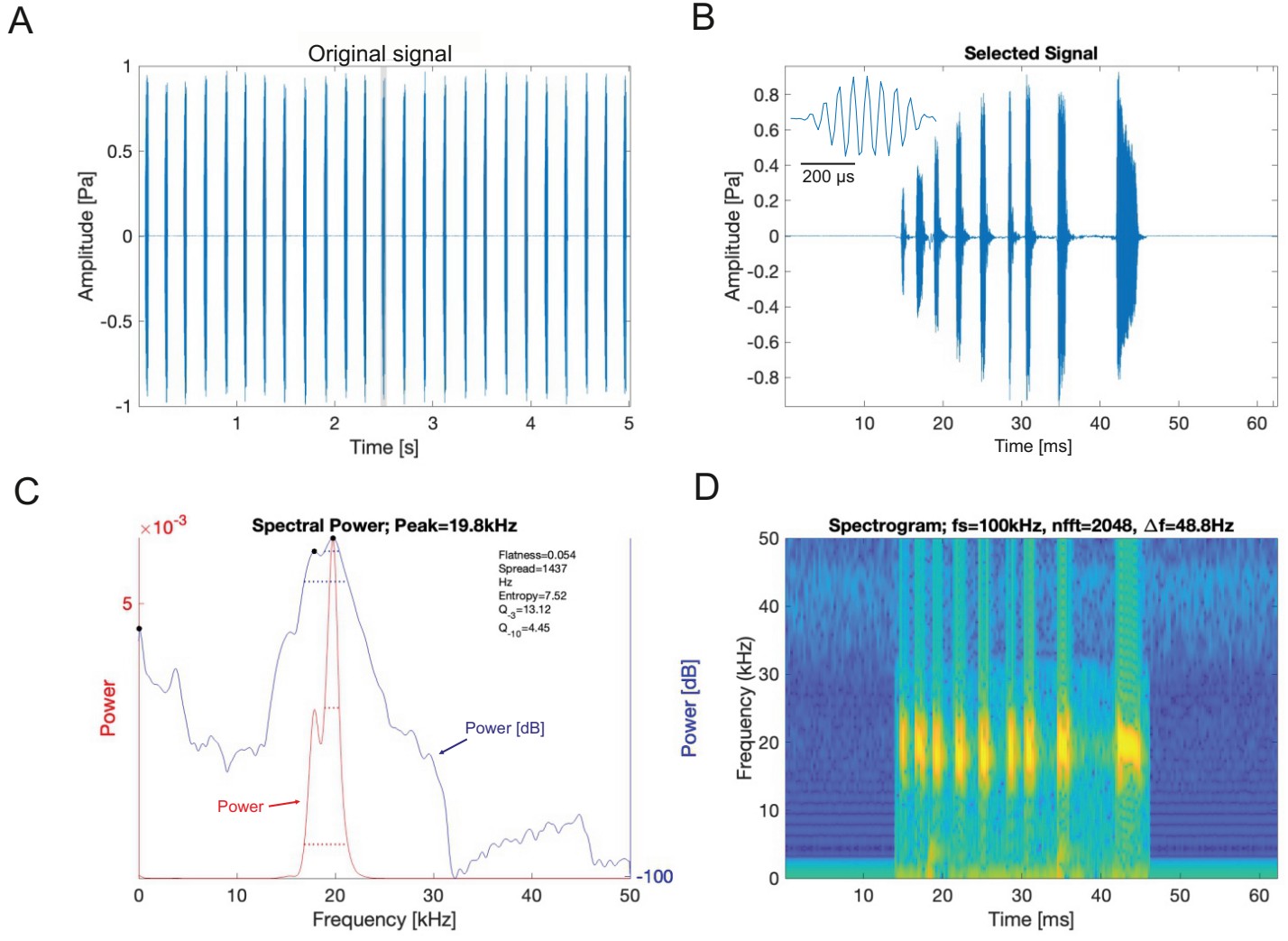

**Figure 11 S. jorgevargasi male calling song under laboratory conditions recorded at a temperature of 19 °C.** (A) A section of the song of *S. jorgevargasi*. (B) Close up view of a syllable showing nine pulses in a sigmoid shape with conserved tonality. (C) Power spectrum of a syllable. (D) Spectrogram of a syllable.     

frequency of *S. huaca* at around 18 kHz as the peak wing resonance was 19.8 kHz (Fig. 16D). The dominant and first vibration mode in the stridulatory areas of the right tegmen is mirrored in the calling song spectrum. As in most Pseudophyllinae, only one resonant area is activated during stridulation, the right mirror (Figs. 14A, 15A, 16A); the left mirror shows an insignificant amplitude of vibration pattern. Under laboratory conditions the calling song duty cycle is relatively constant, with males spending most of the night singing from the burrows they guard and protect.

## Pinnae resonance and numerical models

After 3D printing and stimulating the pinnae cavity with broad frequency sound using scaled wavelengths to match the size of the enlarged 3D models, the peak resonance of each pinnae cavity was revealed along with its acoustic gain (Fig. 17A).

A

**Original Signal**

B

**Selected Signal**

200 µs

C

**Spectral Power; Peak=14.3kHz**

Flatness=0.054
Spread=949 Hz
Entropy=7.13
$Q_{-3}$=11.67
$Q_{-10}$=4.32

Power [dB]

Power

D

**Spectrogram; fs=100kHz, nfft=2048, Δf=48.8Hz**

**Figure 12** *S. sodalis* **male calling song under laboratory conditions recorded at a temperature of 19 °C.** (A) A section of the song of *S. sodalis*. (B) Close up view of a syllable showing 19 pulses in a sigmoid shape with conserved tonality. (C) Power spectrum of a syllable. (D) Spectrogram of a syllable.

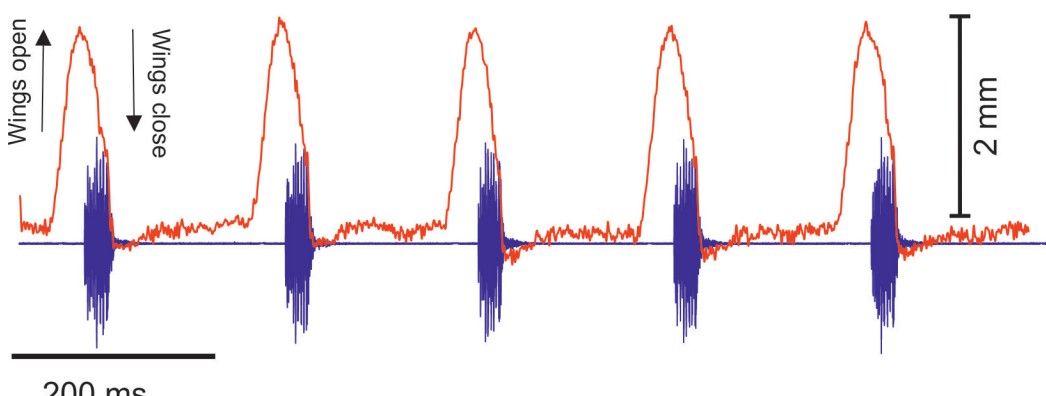

Wings open

Wings close

2 mm

200 ms

**Figure 13 The wing motion of *S. sodalis* (red) overlayed on top of the calling song (purple) recorded at 19 °C.**

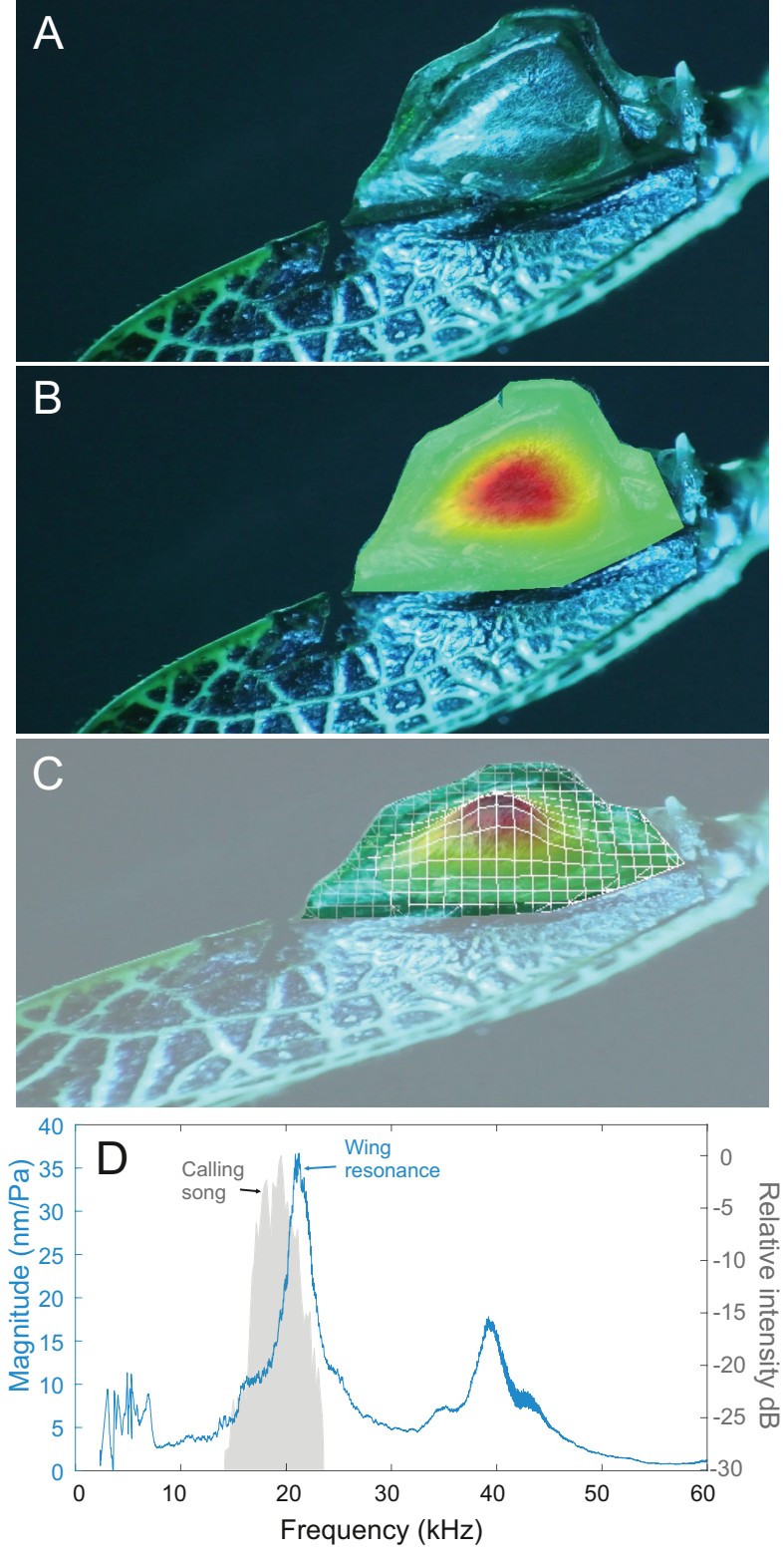

**Figure 14 Wing resonance of *S. jorgevargasi*.** (A) Mirror of the right tegmen. (B) Wing vibration in response to broadband stimulation (periodic chirps from 2–60 kHz). (C) 3D representation of B, showing maximum deformation. (D) FFT of wing vibration showing wing resonance, as compared with call power.

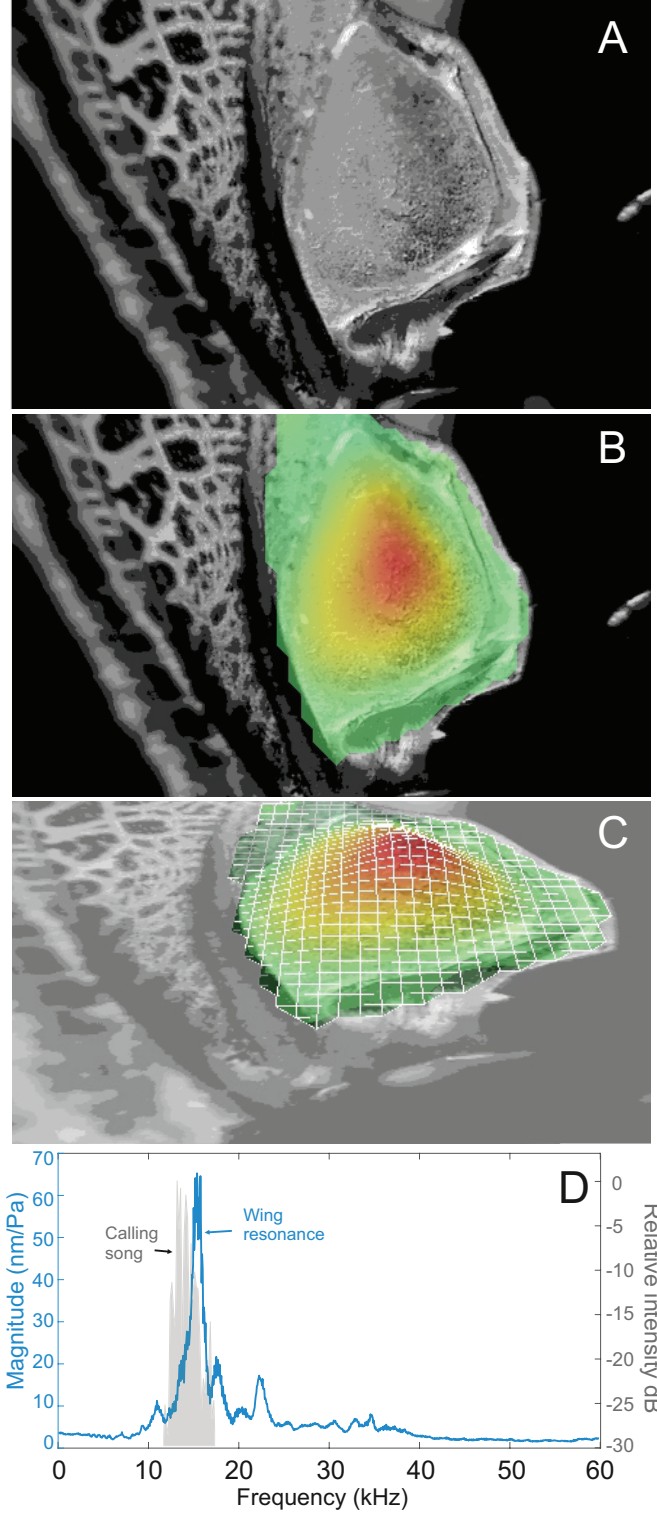

**Figure 15 Wing resonance of *S. sodalis*.** (A) Mirror of the right tegmen. (B) Wing vibration in response to broadband stimulation (periodic chirps from 2–60 kHz). (C) 3D representation of B, showing maximum deformation. (D) FFT of wing vibration showing wing resonance, as compared with call power.

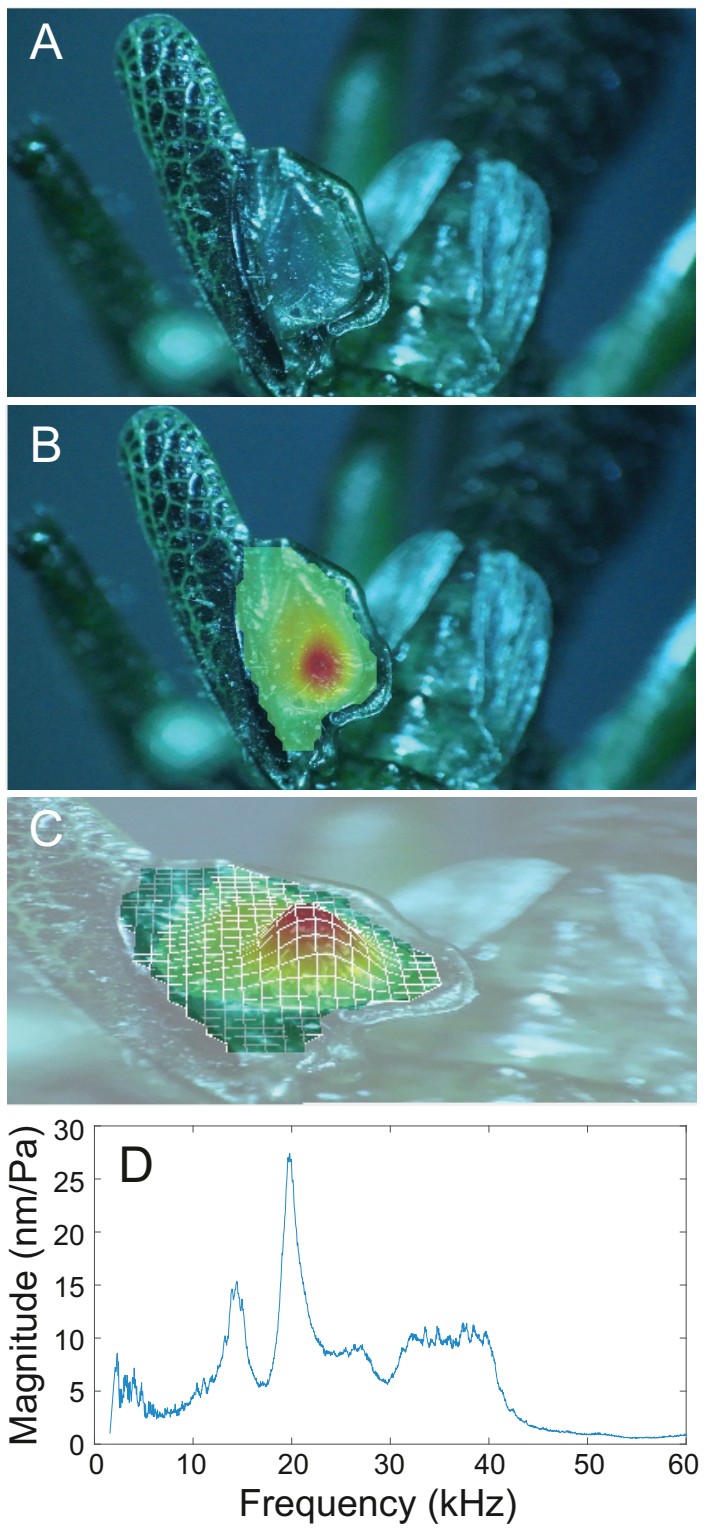

**Figure 16 Wing resonance of _S. huaca._** (A) Mirror of the right tegmen. (B) Wing vibration in response to broadband stimulation (periodic chirps from 2–60 kHz). (C) 3D representation of B, showing maximum deformation. (D) FFT of wing vibration showing wing resonance.

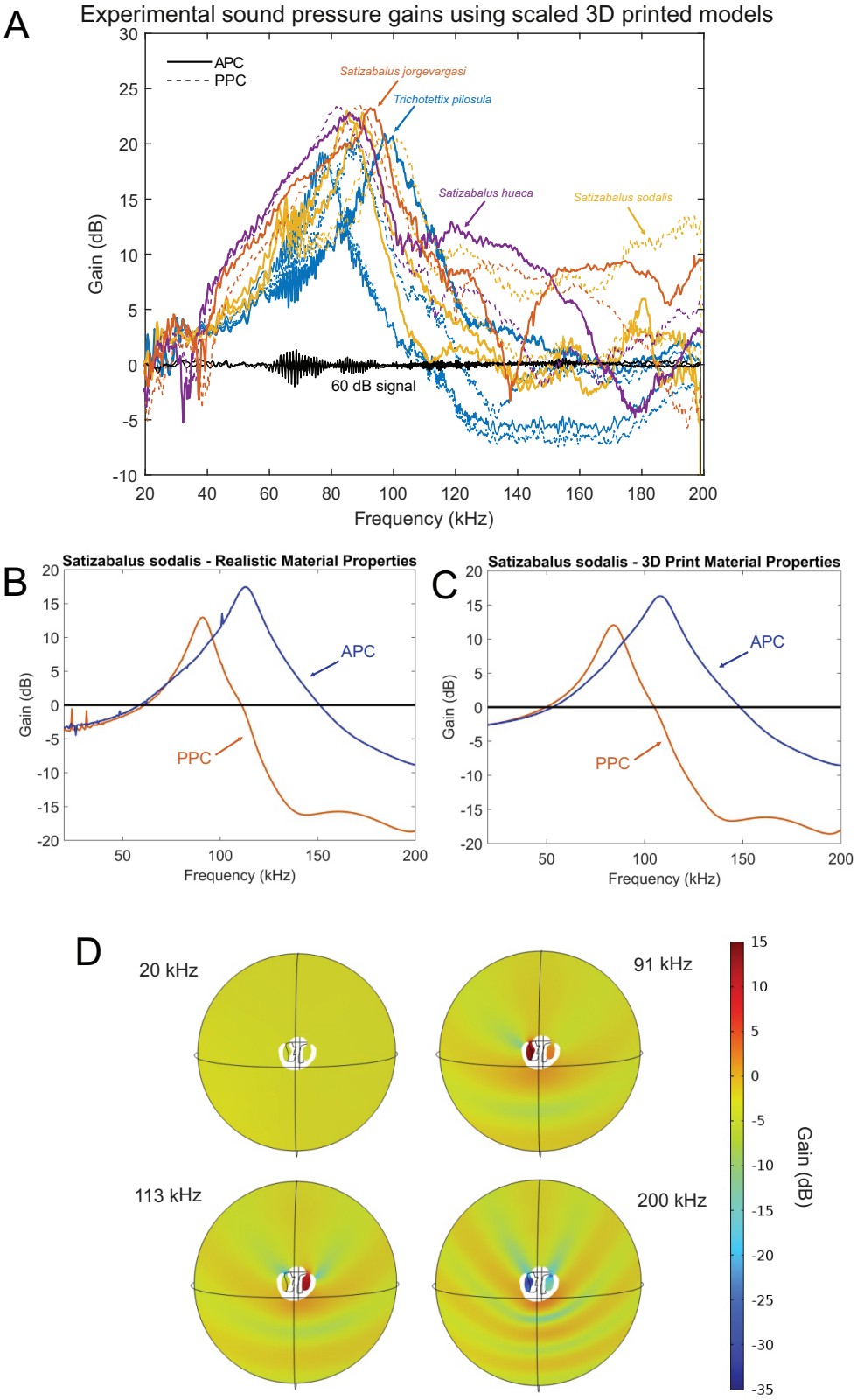

**Figure 17** **Male resonance and acoustic pressure gains in the auditory pinnae of species in the genus *Satizabalus*.** (A) Auditory pinnae resonance of 3D printed models. (B) Pinnae resonance of numerical

**Figure 17** (continued)
model using realistic material properties for *S. sodalis*. (C) Pinnae resonance of numerical model using photopolymer resin material properties for *S. sodalis*. (D) Cavity-induced sound pressure distribution and gains with pinnae. Abbreviations: APC, Anterior Pinnae Cavity. PPC, Posterior Pinnae Cavity.

The average peak resonance of *S. jorgevargasi* was 89.9 kHz with an average acoustic gain of 23 dB. Similarly, *S. sodalis* had an average peak resonance of 91.1 kHz with an average gain of 25 dB. *S. huaca* has a lower average peak resonance when compared to its congenerics at only 78.5 kHz with an average gain of 22 dB (Fig. 17A). For comparisons with a species that shares a similar habitat and suspected phylogenetic relationships to *Satizabalus*, these experiments were repeated in a *Trichotettix pilosula* specimen. In *T. pilosula*, the average peak pinnae resonance was 91.2 kHz with a gain of 24 dB (Fig. 17A).

As a validation of the 3D print pinnae resonances obtained, the numerical simulation of the pinnae acoustic properties was considered in the frequency domain, using the actual size and geometry of the cavities of *S. sodalis*. Two sets of simulations were carried out, where the first one incorporated realistic material properties of *S. sodalis* pinnae and tympana, and the second simulation incorporated the material properties grey ABS-like photopolymer resin for the whole cavity geometry, as used during 3D printing.

Having the realistic material properties of *S. sodalis* pinnae and tympana incorporated into the model gave an average peak resonant frequency of 102 kHz with an average acoustic gain of 15.4046 dB (Fig. 17B). Replacing the material properties with that of grey ABS-like photopolymer resin for the whole cavity geometry returned an average peak resonant frequency of 96 kHz with an average acoustic gain of 14.167 dB (Fig. 17C).

### Field recordings of bats

Between 19.00–20.00 on the 13/02/24, we recorded six echolocating bat calls from three species. Two of the recordings were identified as *Lasiurus blosevillii* with a frequency range of ~40–70 kHz. Three of the recordings belong to *Myotis* cf. *riparius* and have a frequency range of ~60–120 kHz (Fig. 18). The final recording belongs to an unidentified species with a frequency range of 46–52 kHz.

## DISCUSSION

### Bioacoustics

The call characters of *Satizabalus* species are consistent with the range of frequencies exhibited by Cocconotini species, which are normally between 10 and 35 kHz (*ter Hofstede et al., 2020*; *Montealegre-Z & Morris, 1999*; *Montealegre-Z et al., 2017*; *Stumpner et al., 2013*). As well as this, many members of Cocconotini have a song structure that consists of discrete syllables sustained for a few milliseconds, followed by a silent interval which is matched by the songs of *Satizabalus* (*Belwood & Morris, 1987*; *ter Hofstede et al., 2020*; *Montealegre-Z & Morris, 1999*; *Morris & Beier, 1982*; *Morris et al., 1994*). They also manage to conserve purity in their calls despite having discrete pulses and silent intervals, the introduction of silent intervals can increase a noisy tendency in the syllable however, the discrete pulses can be so

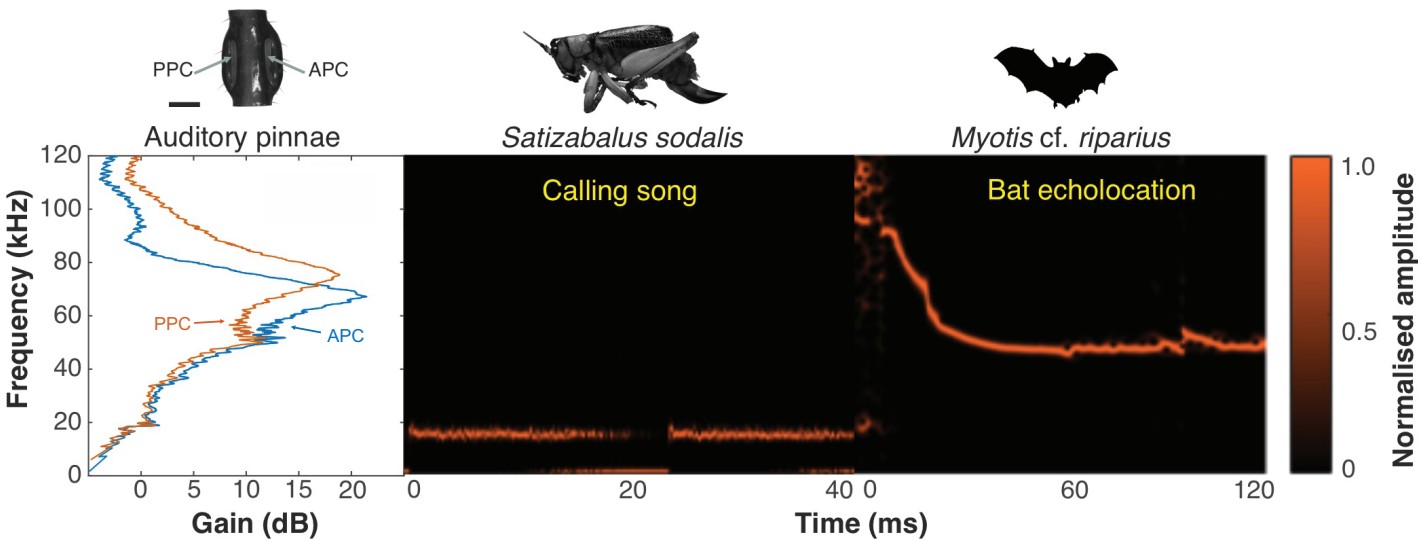

**Figure 18 Ecological relevance of pinnae in *Satizabalus sodalis*.** Auditory pinnae show resonance frequencies above ca 40 kHz, with maximum resonant peaks at around 60–80 kHz (left panel), covering the range of echolocation frequencies of a native insectivorous gleaning bat species, *Myotis* cf. *riparius* (right panel). The conspecific call of *S. sodalis* peaks at 15 kHz and is not enhanced by the presence of the pinnae (central panel). Spectrogram parameters: FFT size 4,096, Hamming window, 128 rectangles, 120 frames, and 93.75% overlap; frequency resolution: 1,024 Hz. APC, anterior pinnal cavity, PPC, posterior pinnal cavity. Silhouette of bat created by Charlie Woodrow using Adobe Photoshop. Photo credit: habitus of female *S. sodalis* and auditory pinnae, Lewis B. Holmes.          

pure that they negate the noise introduced by the silent intervals (*Montealegre-Z & Morris, 1999*; *Montealegre-Z, Morris & Mason, 2006*; *Stumpner et al., 2013*). This call character is displayed by many Cocconotini species and is thought to have evolved as a mechanism to avoid eavesdropping by bats (*Heller, 1995*).

In the laboratory, at 19 °C, *S. jorgevargasi* produced between 4–5 syllables per second with each consisting of 9–10 rapid pulses with a sinusoidal nature tendency (Fig. 11B). The call characters are similar to those displayed by *Trichotettix pilosula* (compare Fig. 11 to Figure 32 in *Montealegre-Z & Morris, 1999*). *Satizabalus jorgevargasi* exhibits a narrow carrier frequency spectrum with a width of 15–24 kHz at −30 dB (Fig. 11C), *T. pilosula* produces a slightly lower spectrum at −30 dB with a spectral width of 13–22 kHz (Figure 32C in *Montealegre-Z & Morris, 1999*). The similarities in the carrier frequency spectra are likely a result of comparable stridulum and similarities in the structure of the main resonant areas of the tegmina. *Satizabalus jorgevargasi* has a greater file length, tooth number, and greater tooth density than *T. pilosula* which would explain the slightly higher carrier frequency (compare Table 1 and Fig. 3 to Table 16 and Figure 20 in *Montealegre-Z & Morris, 1999*).

Similar song recordings of *S. sodalis* in the lab at 19 °C, reveal that the species produces between 4–5 syllables per second with each consisting of 18–20 pulses with a sinusoidal nature tendency (Fig. 12B). *Satizabalus sodalis* is more similar, based upon acoustic characters, to *Trichotettix pilosula* (Cocconotini), supporting its move from *Gnathoclita* (Eucocconotini) to our new proposed genus; *Satizabalus* (Cocconotini). *Satizabalus sodalis* has a carrier frequency peak of 13.4 kHz (Fig. 12C), whilst *T. pilosula* produces a slightly higher carrier frequency peak of 16.3 kHz (Figure 32C in *Montealegre-Z & Morris, 1999*).

The type species of *Gnathoclita*, *G. vorax*, has a very low carrier frequency peak of 8.75– 8.9 kHz at 26 °C (*Hugel, 2019*), which is almost half that produced by *S. sodalis* and highlights the vast difference between the two species.

## Pinnae resonances

Using 3D models, the acoustic properties of the auditory pinnae were measured. As predicted, all three species within *Satizabalus* share a similar average peak resonance between 78.5 and 91.1 kHz and the ability to amplify ultrasonic frequencies between ~40 and 120 kHz by up to 25 dB (Fig. 17A). This is also seen in *Trichotettix pilosula*, a species that shares morphological and behavioural characters with *Satizabalus* spp. along with a similar habitat. *Trichotettix pilosula* had an average peak pinnae resonance of 91.2 kHz and an acoustic gain of 24 dB for that frequency. To validate the use of 3D printed models in place of a real pinnae, two numerical models were created for *S. sodalis*. One tested the resonant frequencies of the pinnae cavities with realistic material properties including the elasticity of the tympanal membranes. The second used the material properties of grey ABS-like photopolymer resin, which is the material used for 3D printing experiments. We found that there was only a 6 kHz difference between the resonant frequencies of the two models with a difference in acoustic gain of 1.2376 dB (Figs. 17B, 17C). This is likely a result of the material properties of the photopolymer resin being very similar to those observed in the insect cuticle, thus proving the reliability of the 3D printed models. The models also show that the pinnae do not show particular resonances for frequencies close to that of the specific carrier frequency (Fig. 17B). These results are in agreement with previous results in the literature (*Pulver et al., 2022*; *Woodrow & Montealegre-Z, 2023*). *Pulver et al. (2022)*, showed that the auditory pinnae in the model species *Copiphora gorgonensis* function to increase the gain of high ultrasonic frequencies above 60 kHz, with the greatest pressure gains seen in frequencies over 100 kHz. They go on to suggest that this serves to enhance the detection of echolocating bats. Similarly, *Woodrow & Montealegre-Z (2023)*, showed that the pinnae of another Pseudophyllinae species of the genus *Eubliastes* also resonates at ultrasonic frequencies beyond the specific carrier frequency.

*Satizabalus jorgevargasi*, *S. sodalis* and *T. pilosula* all have a common average resonance around 90 kHz, with *S. huaca* having a slightly lower average of 78.5 kHz (Fig. 17A). The similarities in the peak resonances could be explained by the similar morphologies of the pinnae (Figs. 2D, 6D, 9C) and possibly the predators that the four species share. *Satizabalus* spp. are all nocturnal and are primarily found in the undergrowth with males calling from the burrows they hide in. Since the males often remain stationary whilst stridulating, it is the female that actively exposes herself to locate the male putting her at greater risk of being predated. Perhaps it is during these periods of vulnerability that the auditory pinnae's ability to detect high ultrasonic frequencies is most prevalent, allowing females to detect and react to bats before being predated. Due to the similar habitats that the four species share, it is expected that they are all predated by similar species of tropical bats and that the pinnae have evolved to best detect those specific frequencies that they produce. This has been shown in some species of noctuid moths (Noctuoidea) where their

hearing is tuned to detect the calls of local bat species (*Fullard, 1984a*), and it is well known that katydids are capable of detecting the calls of echolocating bats despite the presence of background noise (*Hartbauer, Radspieler & Römer, 2010*). Furthermore, katydids will cease calling upon detecting the calls of gleaning bats (*ter Hofstede & Fullard, 2008*; *ter Hofstede, Ratcliffe & Fullard, 2008*) proving they are capable of detecting and reacting to these ultrasonic frequencies. A preliminary survey of echolocating bats from the same locality as *Satizabalus sodalis* revealed that the insectivorous bat, *Myotis* cf. *riparius*, is common in this area of the Andes and produces ultrasonic sweeps from ~120 to 60 kHz (Fig. 18) (*Arévalo-Cortés et al., 2024*). However, our recordings of this species show that the sweep starts at around 90 kHz (Fig. 18). The maximum resonance peaks of the auditory pinnae in *S. sodalis* are between 60 and 80 kHz, within the range to detect the earliest part of the echolocation sweep of *M. riparius* (Fig. 18). So, it is possible that the peak resonances of the pinnae are adapted to better detect the local bat species, however a more in-depth survey of the local echolocating bat species would be beneficial to advance this hypothesis.

## Morphology

The unusual cephalic development seen in the males of *Satizabalus* (Figs. 2A, 6A, 9A) has resulted in largely exaggerated features in the mandibles and rostrum that could be considered as weaponry. Weaponry is rare within Orthoptera, with examples being seen in five species of the family Anostostomatidae (*Field & Deans, 2001*); the New Zealand tusked wētā (*Trewick & Morgan-Richards, 2004*); two South African species, *Libanasidus vittatus* (*Kirby, 1899*) and *Libanasa capicol* (*Peringuey, 1916*); three species in the genus *Listrocelis Serville, 1831* (*Fialho et al., 2014*); and two species from the genus *Dicranostomus* Dohrn, 1888 (*Heller & Helb, 2021*). The genus *Gnathoclita Haan, 1843* (Eucocconotini), has comparable mandibular morphology to that seen in *Satizabalus* (*Hugel, 2019*; *Montealegre-Z & Morris, 1999*). However, aside from the facial features, there are no other similarities that are shared. The type species, *G. vorax*, along with *G. laevifrons, G. izerskyi*, and *G. peruviana* are all much larger in overall body size than *Satizabalus* and possess tegmina that cover the entire length of the abdomen (*Cigliano et al., 2024*). The morphology of the mirror differs between the two genera, in *Gnathoclita* species the mirror is elongated and oval shaped, whereas the mirror of species in the genus *Satizabalus* is much more circular and has a tear drop shape (compare Figs. 3A; 7A; 10 to 7D in *Hugel, 2019*). Additionally, the male cerci are unrecognisable (compare Figs. 2G; 6G; 9E to Figs. 33; 36 in *Gorochov, 2018*). Geographical distribution separates the two genera. The *Gnathoclita* species are all found in the Amazon regions in Peru and Brazil (*Hugel, 2019*) as well as Suriname, Guyana, and French Guiana (*Cigliano et al., 2024*), making them a lowlands species and fully separated from the Colombian Andes. *Satizabalus* is only found in the cloud forests of Colombia; *S. jorgevargasi* from the Western slope of the Central Cordillera montane forest, *S. sodalis* from the Eastern slope of the Western Cordillera, and *S. huaca* from the Nudo de los Pastos. *Satizabalus sodalis* shares more morphological similarities with *Satizabalus* than it does with *Gnathoclita* (compare Fig. 2 and Table 1 to Fig. 14 and Table 13 in *Montealegre-Z & Morris, 1999*) and is separated geographically from all other species within *Gnathoclita*.
*Satizabalus* is comparable in morphology to species in the genus *Trichotettix Stål, 1873* (Cocconotini), most so to *T. pilosula*, and an undescribed species of the same genus, both of which also inhabit the cloud forest of the Andean forest in Colombia. They both share similar morphological measurements (compare Table 1 to Table 2 in *Montealegre-Z & Morris, 1999*) and forms (compare Figs. 2 to 3 in *Montealegre-Z & Morris, 1999*). However, males in the genus *Trichotettix* do lack the enlarged mandibles seen in *Satizabalus*. Coupled with similarities in acoustic data, it would be worth further investigating the phylogenetic relationships of these two genera.

## Ecology and behaviour

In the wild and in captivity, males will stridulate from a burrow that provides them protection from predators. These burrows are typically the hollow stems of dead plants, although in captivity they will create a shallow burrow, no more than 5 cm in depth, in the sphagnum moss substrate. Females will detect these calls and actively seek the male, during which they make themselves vulnerable to predation. It has previously been shown that the gleaning bat, *Micronycteris microtis*, prefers moving prey (*Geipel et al., 2020*). Once females find a singing male, they will proceed to enter the burrow and copulate with the male. Females will then remain at the base of the burrow for some time, perhaps because the burrow offers them some protection from predators, but how long they remain in the burrow is unknown.

The enlarged mandibles seen in *Satizabalus* males (Figs. 2A, 6A and 9A) could be an adaption for aggressive behaviours. Montealegre-Z and Morris suggest that the previously named *G. sodalis*, may utilise these for aggressive behaviours similar to those seen in the Australian tree wētā (*Montealegre-Z & Morris, 1999*) which exhibit male-male aggression (*Brown & Gwynne, 1997*). However, there are no confirmed reports of this behaviour being witnessed within *Satizabalus* or *Gnathoclita* and so their function remains unclear. In captivity, we have observed males chasing one another around the enclosure whilst stridulating but never have there been any physical fights witnessed.

Mate guarding behaviour is exceptionally rare within Tettigoniidae with a few instances being recorded within *Oncodopus* and *Colossopus* (*Ünal & Beccaloni, 2017*). It has also been reported by Hugel to occur within *Gnathoclita* (*Hugel, 2019*). *Gnathoclita vorax* was commonly found stridulating from hollow tube-like plant sections that are only just wider than the male's head, allowing them to almost completely block the entrance (*Hugel, 2019*; *Naskrecki, 2008*). In three cases males that were found stridulating were alone within their structures. However, two males that were not stridulating were found blocking the entrance to their structure, and when they were removed it was found that there was a female at the base of the structure, although without a spermatophore attached (*Hugel, 2019*). These observations strongly suggest that *G. vorax* males exhibit some form of mate guarding behaviour (*Hugel, 2019*). This behaviour parallels what we have observed within *Satizabalus*. Both in captivity and in the wild, we have observed males position their heads towards the entrance of a burrow and begin to stridulate, and males will cease to sing once a female is at the base of a burrow. The following is anecdotal; a strange occurrence was observed in the field during a field trip to Colombia. A male *S. sodalis* was found guarding

the entrance to a burrow, once he was removed it was revealed that there were two females at the base. One was alive and well, whilst the other appeared to be paralysed and attempts to revive this female were unsuccessful as she died a few days later. The reason for this remains unclear.

The striking colouration seen in the tegmina and legs of male *Satizabalus* species (Figs. 4, 8) could have an aposematic function. This is seen in some katydids including *Acripeza reticulata* (*Umbers & Mappes, 2015*) and the Crayola katydids, *Vestria* spp. (*Nickle et al., 1996*), however these colours are hidden until threatened at which point, they are boldly displayed. Members of *Satizabalus* make no attempt to hide their colouration. It may also serve to convey fitness to a potential mate or rival. However, being a nocturnal species, conveying sight-based signals (colourful structures) may not be effective, but UV components from moonlight may provide visual cues. Further investigation is required to confirm the function of the colouration seen within *Satizabalus*.

## CONCLUSIONS

There are 36 described genera in the tribe Cocconotini (*Cigliano et al., 2024*). *Satizabalus* is hereby described as a new polytypic genus to science based upon the morphological and acoustic evidence presented. The type species being *S. jorgevargasi* (Figs. 2, 4) and a second new species being *S. huaca* (Fig. 9). Additionally, we propose that *G. sodalis* (Figs. 6, 8), currently positioned within *Gnathoclita* (Eucocconotini), should be assigned to *Satizabalus* (Cocconotini) based on the acoustic, morphological, and behavioural data provided within this report. The three species here described all produce differing calling songs with variation in their call characters and carrier frequency (Figs. 11, 12), however they are all consistent and within the range exhibited by Cocconotini species. As in most species of Pseudophyllinae, only the right mirror is active during stridulation. Additionally, there is also a clear relationship between the wing resonance and calling song carrier frequency in *Satizabalus jorgevargasi* and *S. sodalis* (Figs. 14, 15) which allows us to predict the carrier frequency of *S. huaca* (Fig. 16). Lastly, the auditory pinnae of the three species along with those of *Trichotettix pilosula* have a comparable peak resonance and all increase the acoustic gain of high ultrasonic frequencies between ~40 and 120 kHz by up to 25 dB (Fig. 17A), further supporting their use as ultrasonic detectors.

## ACKNOWLEDGEMENTS

This work is dedicated to Glenn Morris, who funded and participated in pioneering expeditions to Bosque el El Ensueño (Valle del Cauca), and Reserva La Planada (Nariño). We thank Holger Braun for feedback on this research presented as a poster during the Invertebrate Sound and Vibration conference 2023. We also thank Dr Janse Bittner, Pedro Moreno, and Maria Fernanda Moreno who provided permits and accommodation enabling work at the Reserva La Planada in 2001. Abelardo Nastacuaz and Lehyla Patricia Moncayo helped during fieldwork at this location.

### Funding

This research was founded by an NSF - NERC grant NSF DEB- 1937815 - NE/T014806/1 (to Fernando Montealegre-Z), and by a European Research Council Grant ERCCoG-2017-773067 (to Fernando Montealegre-Z for the project "The Insect Cochlea"). The funders had no role in study design, data collection and analysis, decision to publish, or preparation of the manuscript.

### Grant Disclosures

The following grant information was disclosed by the authors:
NSF - NERC grant NSF DEB-1937815-NE/T014806/1.
European Research Council: RCCoG-2017-773067.

### Competing Interests

The authors declare that they have no competing interests.

### Author Contributions

- Lewis B. Holmes performed the experiments, analyzed the data, prepared figures and/or tables, authored or reviewed drafts of the article, and approved the final draft.
- Charlie Woodrow conceived and designed the experiments, performed the experiments, analyzed the data, prepared figures and/or tables, authored or reviewed drafts of the article, and approved the final draft.
- Fabio A. Sarria-S performed the experiments, analyzed the data, authored or reviewed drafts of the article, and approved the final draft.
- Emine Celiker conceived and designed the experiments, performed the experiments, analyzed the data, prepared figures and/or tables, authored or reviewed drafts of the article, and approved the final draft.
- Fernando Montealegre-Z conceived and designed the experiments, performed the experiments, analyzed the data, prepared figures and/or tables, authored or reviewed drafts of the article, and approved the final draft.

### Field Study Permissions

The following information was supplied relating to field study approvals (*i.e.*, approving body and any reference numbers):

The Colombian Ministry of Environment and Sustainable Development approved permits to conduct fieldwork in Colombian national parks (decree 309 of the 25 February 2000; and DTS0-G-090 14/08/2014). Collected specimens were transported to the University of Lincoln, UK, under collection and exportation permit Expedient PIDB DIG No. 0009 - 14, auto numero 108 03 Junio de 2014 and No. 00645 09/24/15 (issued by the Colombian Authority of Environment Licences (ANLA)).

## Data Availability

The data is available at FigShare: Holmes, Lewis; Woodrow, Charlie; Sarria, Fabio; Celiker, Emine; Montealegre-Z, Fernando (2024). Wing mechanics and acoustic communication of a new genus of sylvan katydid (Orthoptera: Tettigoniidae: Pseudophyllinae) from the Central Cordillera cloud forest of Colombia. figshare. Dataset. https://doi.org/10.6084/m9.figshare.25196771.v4.

## New Species Registration

The following information was supplied regarding the registration of a newly described species:

Publication LSID: urn:lsid:zoobank.org:pub:0CC2E5D2-6B00-4186-83FD-380790400A78

Satizabalus Genus LSID: urn:lsid:zoobank.org:act:EFD7AFFC-818F-4BBA-BE17-E651720A0042

Satizabalus huaca species LSID: urn:lsid:zoobank.org:act:D4C70254-E60C-4208-B0AD-9218433FB741

Satizabalus jorgevargasi species LSID: urn:lsid:zoobank.org:act:51191C36-378B-4FB8-885E-0F9B00048B7C

Satizabalus sodalis species LSID: urn:lsid:zoobank.org:act:8FDAA70F-F5B4-4A25-9C0F-282D4863C56B.

## Supplemental Information

Supplemental information for this article can be found online at http://dx.doi.org/10.7717/peerj.17501#supplemental-information.

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
