# Peer review of "Wing mechanics and acoustic communication of a new genus of sylvan katydid (Orthoptera: Tettigoniidae: Pseudophyllinae) from the Central Cordillera cloud forest of Colombia"

_PeerJ, doi:10.7717/peerj.17501_

## Round 0.1 · original submission · Major Revisions

All three reviewers suggested major revisions. Please, consider their comments and provide point-by-point response, when resubmitting new version of your manuscript

·

Basic reporting

See below (4)

Experimental design

See below (4)

Validity of the findings

See below (4)

Additional comments

The very interesting paper "Wing mechanics, hearing and bioacoustics of a new genus of sylvan katydid (Orthoptera: Tettigoniidae:  Pseudophyllinae) …" consists of two relatively independent parts. In the first one wing mechanics, hearing and bioacoustics are treated, while the second is devoted to taxonomy s.l. For the first I have only few suggestions for improvement (see below), but the second presents several serious problems.
To start with a more technical one
-Please follow the journal's new species policies (https://peerj.com/about/policies-and-procedures/#new-species). Most important: The authors will need to contact Zoobank to obtain LSIDs for all new taxa.
The next problem are the diagnoses.
Although in many paper the diagnoses have a poor quaiity (as e.g. in Cadena-Castañeda et al. 2020 and here), the ICZN says clearly (glossary):
"diagnosis, n.
A statement in words that purports to give those characters which differentiate the taxon from other taxa with which it is likely to be confused." (= differential diagnosis)
While it should be no problem to change the species' diagnoses accordingly. in the case of the new genus it is more difficult, especially as species from two tribes are combined. At first the differences between Cocconotini and Eucocconotini have to be discussed, then the placement of the new genus decided and finally in the diagnosis the differences to the other genera of the tribe discussed, possibly somewhere also refuting Gorochov's (2018) view of Gnathoclita (including Disceratus)? (May be something like in Cadena-Castañeda et al. 2020; Cocconotini paraphyletic according Mugleston et al. 2018)
Another problem is the key (see below) and finally the invalid selection of new types for G. sodalis.

Some minor or specific comments/problems

l 37 Satizabalus -> Satizabalus n.gen.
l 38/39 (S.  sodalis n. comb.) and two new species (S. jorgevargasi n. sp. and S. hauca n. sp). It is helpful for the reader if n.gen and n. sp. are always used when using the respective names. (or sp. nov., gen. nov., comb. nov.)
l 57 Pseudophyllinae: Cocconotini -> Pseudophyllinae, e.g. Cocconotini
l 79 and a single G. sodalis - 'single' necessary? Number of five already completed
l 82 The sister tribe, -> A related tribe,
l 83-5 Please explain why Trichotettix may be similar/related to Gnathoclita
l 89 in a hereby newly described -> Something like: Using these data we will demonstrate that the Colombian specimens represent a new Cocconotini genus.
l 97 Coccononotini -> Cocconotini
l 134 tegmina -> tegmen; S.  sodalis, and S. huaca - check italics
l 136 Waldronn - > Waldbronn
l 136 According to line 134: The right tegmina were -> The right tegmen was. See also l 143
l 162 S. sodalis Comb. nov -> S. sodalis comb. nov 
l 200 Authors' names not necessary but if given please give the names of all authors
l 206 Please add: Included species Satizabalus huaca n.sp., here described., S. sodalis n. comb.
 l 207- Diagnosis unsatisfactory. A diagnosis of a new genus must at first describe why the genus belongs to the next higher group (Cocconotini) and then how it differs from the other genera of that tribe. The present text belongs to description.
l 225 Key unsatisfactory.
#1 Please give colours of palps and eyes for all three species.
#2 Please give absolute dimensions - relations do not help. Body size xx-xx in huaca, yy-yy in sodalis. Give data for "Male tegmen to body length ratio" in huaca. Omit here all references to colour (either identification identical in both or known only in one species)

l 237 Please replace "N" with "n" everywhere (always only samples sizes used; see enclosed pdf)
l 230 1'. Labial pals -> 1'. Labial palps
l 239  soldalis -> sodalis 
l 245 Costa Rica: Where is this place - department, co-ordinates, altitude?
l 246/7 Costa Rica - Why now in Valle del Cauca ?
l 255-9 Diagnosis unsatisfactory. In the diagnosis differences to congeneric are to be described (e.g. "Eyes are primrose yellow in life in jorgevargasi " while they are xx in huaca and yy in sodalis). Transfer non-diagnostic details to description.
l 312 Satizabalus sodalis Comb. Nov-> Satizabalus sodalis (Brunner von Wattenwyl, 1895) n. comb.
next line:
Gnathoclita sodalis Brunner von Wattenwyl, 1895 Verh. der Zoologisch-Botanischen Gesellsch. Wien 45:179
l 315-320 Delete your text
Holotype male, type locality unknown, depositary NMW Vienna Museum (male body length 25 mm, in Hebard 26.4 mm, in Beier 1960 20-26 mm and also larger than 15 mm in the photos on OSF).
Using "The whole-body measurement is taken from the most proximal region of the pronotum to the most distal end of the last tergite" makes comparisons with literature data impossible! Your measurements must not be called body length (as in Table 1), but thorax-abdomen-length. This thorax-abdomen-length may be measured for the type on the photos in OSF.
By transferring a species into a new genus the type material does not change!!
l 322-24 Delete your text, list your data under distribution (together with other literature data and from OSF) and explain why you think that your specimens belong to this species.
l 332 Description -> Re-description based on living specimen
l 380 Where is the second male?
l 425 the right mirror (Fig. 14A); Fig. 15A); Fig. 16A); - Check brackets
l 427 How do these burrows look like and are made of?
l 474-80 Disceratus nubiger Eucocconotini 19 kHz similar song structure
vorax much larger (body length 40-60 mm; De Jong 1971) see l 537-9, lower frequency expected
517 Please present evidence that Myotis riparius is a gleaning bat contrasting to "Its diet includes a large variety of insects, particularly those in the orders Coleoptera, Diptera, Lepidoptera, and Orthoptera, all caught in flight (LaVal and Rodríguez 2002)." from GBIF
What about the calls of gleaning and whispering Micronycteris?
l 626 relocated??
l 636 buy-> by / by up?
Fig. 10 (A) A section of the song of S. sodalist. -> (A) A section of the song of S. sodalis.

·

Basic reporting

(see additional comments below)

Experimental design

(see additional comments below)

Validity of the findings

(see additional comments below)

Additional comments

Comments and suggestions according to line numbers:
33 most tettigoniids / most Tettigoniidae
48 Male katydids
70 detection of ultrasonic predators
77-82 (needs rewording, after “five known species” are mentioned only three species in parenthesis, before “most of them described from Peru” are mentioned four species of which only two are described from Peru)
80 this latter species / the latter species? (depending on new preceding sentence)
80-81 (significantly different in size – much smaller?)
81 (there are only two “Peruvian congenerics”)
89 also containing?
90 katydids
90 montane forest (but “west and central cordilleras” could perhaps be capitalized as geographical names)
91 bioacoustic data (adjective like morphological, geographical)
93 new genus Satizabalus
111 (Colombian environment Ministry - use fully capitalized official English name?)
122 (what does NI mean?)
124 20 cm
134 single right tegmen
134 a male of each
142 15 cm
146 Vereda Chicoral, La Cumbre
147 Echo Meter 2 (proper name)
147 Wildlife Acoustics
148 Echo Meter App (Wildlife Acoustics …. (move “for iOS” in front of the parentheses)
150/151 m a.s.l. / MASL
159 Sigma Corporation
162 comb. nov. (“n.sp.” should probably already be used at earlier mentions of the new names)
165/166 Comet Technologies
176 which was placed?
185 Comet Technologies
200 (maybe don’t mention any author here, then it would be clearly Satizabalus Holmes, Woodrow, Sarria-S & Montealegre-Z, 2024 – but perhaps four author names for genus and species are exaggerrated and could be reduced to the author(s) who examined the speciemes and wrote the descriptions)
204 Western Cordillera
206 Type species
207-211 (include a few more really diagnostic features, especially those that place the genus in Cocconotini and distinguish it from similar genera)
209 curved / curving
209 (if this mandible to head ratio is not explained it is not very useful for a genus diagnosis)
211 (varying tooth number is practically useless for genus diagnosis)
213 (oval is a two-dimensional shape, from what viewing angle is the head oval?)
213 (the rostrum cannot overlap with itself, apparently the mandibles are meant)
214 (varying colour is no description, the different colors need to be explicitly mentioned)
215 (almost twice the length of body would be very short for Pseudophyllinae, in Fig. 1 the antennae are not fully shown but at least twice as long as the body, I guess they are much longer)
216 (what colour is gambogo yellow?)
216 (what means “matching colouration”?)
218 (“wider at the base before tapering distally” is true for all tettigoniids, a description needs to be more specific)
219 (the pinnae are always clearly visible in pseudophyllines)
223 (vatiation within or between sepcies?)
227/228 Central Cordillera
230 Western Cordillera
232 (“compared to congenerics” is useless in a key, you need a precise range of measurements to identify a particular species)
233 (how long are the tegmina in relation to the pronotum?)
236 (how long are the tegmina in relation to pronotum or abdomen?)
237/238 (these ratios are strange, in males the relationship of tegmen length to body length seems to be about 0.3, better set in relationship to pronotum or abdomen length whithout any numbers)
243 Figs. 1, 3
254 Central Cordillera
256 (“where they come together” – do you mean apically / at the tip?)
257 (“more open than congenerics” – what does that mean, are the ear openings wider? describe state in this very species)
258/259 (in the male the ratio is 0.59, not 0.59:1, the latter for me would mean that tegmina are 59% the length of the body which is wrong; I think these numbers are not very useful and should be replaced or at least supplemented by more straightforward information, like “tegmina almost twice as long as pronotum” or “hardly covering half of the abdomen” or somthing similar)
262 (oval in frontal view?)
264 (antenna are surely longer than twice the body in unharmed individuals)
265 (what means “dorsally compressed”?)
270-271 (”forward facing” would mean that they are directed to the frontal end of the insect, and forward-facing as an ajective is probably hyphenated)
273 (see previous comments on counterintuitive numbers)
274 7.01 mm and area of 3.39 mm2
277 (female cerci are usually always externally visible in pseudophyllines, even in long-winged species).
278 (“as you progress distally” sounds strange, like being a flea walking along the ovipositor)
279 (“one per lobe” is redundant, if styli are present there are always two)
284 Dutch orange (not all entomologists know what type of orange this is)
286 (“gambogo” would be unknown to the great majority of the readers)
296 is currently
302 growing, taking
303 6 ms
310 (what is “r.e.”?)
310 μPa (micropascal is spelled with a lowercase Greek mu)
312 comb. nov
315, 317, 318 Collector (it’s only one in each case)
322 (this is not etymology, etymology would be the explanation where sodalis is derived from or what it means: apparently it is Latin and means companion, alluding to being the second of originally two species in the genus Gnathoclita)
324 eastern slope of the Western Cordillera
325 (most readers will not know what “orpiment orange” is)
326 mandibles apically rounded?
328, 329 (“proportionally longer” – this is uniformative, a species description needs exact information, see previous comments o ratios and numbers)
334 (antennae are much longer, about four times as long as body, see observations of Gnathoclita soldalis on iNaturalist)
344 9.02 mm
345 5.42 mm2
356 mid
357, 358 (again esoteric colour names: gambogo yellow, orpiment orange)
366 is currently unknown
366, 367 mate, however … / mate. However …
369, 370 (it produces a song, and the song shows this structure, but to produce a song structure sounds strange, I think already with the previous species)
376 (“r.e.” is unclear)
376 20 μPa
379 Fig. 12 (reference to figure is missing)
380 Collector
386 the Incas
387 montane cloud forest
388 (mention exact size range, see previous comments on ratios and numbers)
390 (what is the differece between a “narrow opening” and a “tight opening” (previous species)? – a narrow opening is necessary that sound can enter, so that “tight” with the pinnae touching the opposite margins is probably wrong)
392 (see previous comments on ratios and numbers)
395 (“wide oval” in frontal view?)
395 Genae dominate?
396 (antennae must be much longer)
398 (“dorsally compressed” is again unclear, maybe “medially compressed”?)
399 above front coxa (but this is the same in all three species, how narrow is it exactly? if it’s circular than it’s rather small ot tiny than narrow)
400-401 (“Hind femora much wider at base before tapering distally” this general description applies to practically all katydids)
403 (forward-facing is probably not adequate)
403 (if the spines on the hind tibiae were facing forward
403, 404 (hind femora should be mentioned before hind tibia, it needs to be mentioned if they are ventral or dorsal spines, and ventral spines on the hind femur would face backward, to the rear end of the insect).
405 (see previous comments on ratios)
406 8.04 mm and area 4.4 mm2
413 (again esoteric gambogo yellow)
418 … frequency. S. jorgevargasi … (new sentence)
419 (perhaps repeat here the carrier frequency peaks of the two species to understand the prediction with 1.8 kHz difference to wing resonance)
425 (there is one opening parenthesis and three closing parentheses)
433 its acoustic gain?
453 between 10 and 35 kHz?
470, 471 than T. pilosula
480 (“with a sampling rate of 96 kHz” – this means that the carrier frequency spectrum can be recorded up to 48 kHz, but the measured peak at 8.75 – 8.9 kHz is independent of the sampling rate, so it’s probably redundant and confusing to mention it).
485 between 78.5 and 91.1 kHz (a hyphen is usually read as “to”)
486 between ~40 and 120 kHz
487, 488 morphological and behavioural characters (habitat is mentioned at the end of the sentence)
493 pressure gains
515 they are capable
518 between ~ 120 and 60 kHz / from ~ 120 to 60 kHz
520 between 60 and 80 kHz
529 Weaponry is rare
539 (live idividuals of Gnathoclita vorax are also colorful, see observations on iNaturalist).
540 (again esoteric colour name gambogo)
544 The Gnathoclita species
545 (the distribution of G. laevifrons is unknown, G. vorax occurs also in Guyana, Suriname, and French Guiana)
547 Central Cordillera
548 Western Cordillera
554 Andean forest
568 but for how long they remain in the burrow is unknown
575 never have there
578 (Ünal and Beccaloni, 2017)
593 The reason?
595 The striking … (“purpose … could serve” sounds strange)
597 Crayola katydids, Vestria spp.
604 (The Pterochrozinae are currently not a tribe of Pseudophyllinae)
605 (no, they imitate leaves, especially decaying leaves, and disrupting a predator’s search image is completed by intraspecific colour polymorphism, see Braun 2015, Zootaxa 4012, 1-32)
606 (“Although” does not seem to be the right word here)
622 (now it would be Cigliano et al. 2024, the version accessed in April 2023 now has a different URL: http://orthoptera.archive.speciesfile.org)
636 between ~40 and 120 kHz by
636 further supporting?
644 who provided?
784, 787, 791 (these papers are cited as “Hofstede et al.”, here she is listed as “Ter Hofstede”)
799 Ünal M

Reviewer 3 ·

Basic reporting

The article deals with the description of new taxa and introduces nomenclatural changes in the system of South American katydids of the tribe Cocconotini (Tettigoniidae: Pseudophyllinae). In addition, with the help of modern acoustic method, micro-computed tomography and 3D modeling, the sound signals of these insects were recorded and the acoustic resonances of their sound apparatus and one of the parts of the auditory system were studied.
The title of the article contains the term “bioacoustics,” which refers to acoustic signaling. Bioacoustics is the science of acoustic communication, which includes all parts of the corresponding communication system. Main ones are sound production, hearing, communication channel. Therefore, it is better to replace this term in the title with a more precise designation of the subject of research - sound production or acoustic signals.
The structure of the article contains all sections recommended by the editors of PeerJ. However, it is advisable to change the dividing RESULTS into sections , since the existing structure makes it difficult for the reader to compare and analyze the data presented. In particular, the description of the signals of each of the studied species can be transferred from the Taxonomy section to a special section Acoustic signals. Key to species of Satizabalus should be moved to the end of the Taxonomy section.
Figures. Photographs of the morphological characters of katydids (Figs 3, 4B, 8, 9B, 12) are not of high enough quality: they are very dark, and scale bars are almost always absent.
Unfortunately, I did not find any links to additional or raw data that could supplement and/or clarify the main material. An exception is a link to a video from another article of other authors (see L 178).

Experimental design

Comments on sections of the article
ABSTRACT and INTRODUCTION
The INTRODUCTION and ABSTRACT should more clearly justify the choice of the purpose of the study and the need to carry out the stated research.
MATERIALS and METHODS
It is necessary
to provide the main characteristics (frequency range and range of flat response) of microphones and loudspeakers,
describe the method of recording the movements of the tegmina of a singing katydid (see Fig. 11) indicating the equipment used.

When describing experiments with the detection of mirror resonance, it was indicated that the tegmen was stimulated with a broadband sound of 2-60 kHz, but in the captions to Figures 14-16 a pure tone of 60 kHz appears as a stimulating sound. What kind of stimulus was there really in the experience?
Methods for calculating Helmholtz resonance in the ear pinnae cavities are not given.
It is extremely necessary to explain the terms that are used to designate fragments of the temporal pattern of calling song (preferably in a special small section and in accordance with Baker, Chesmore, 2020 (Baker, E., Chesmore, D., 2020. Standardisation of bioacoustic terminology for insects. Biodivers. Data J. 8, E54222. https://doi.org/10.3897/BDJ.8.e54222).

RESULTS and DISCUSSION
When describing new species, data on the holotype, allotype and paratypes is provided. According to the requirements of the ICZN, it is necessary to indicate the storage location of paratypes. The allotype may not be specifically identified, but the corresponding specimen may be added to the paratypes.
The authors a priori believe that the main resonator of the wing during singing is the mirror of the right tegmen, and conduct corresponding experiments, stimulating the dry right single tegmen with sound via loudspeaker. However, the pioneer of these studies, Bailey, showed in 1970 that the main resonator is not the thin membrane of the mirror, but thick veins that form a cell in which this thin membrane is enclosed. At the same time, the resonant properties of the mirror were studied when the stridulatory file hits the plectrum, that is, when two tegmina come into contact during sound production. The induced vibrations of the mirror membrane of the isolated tegmen when stimulated by sounds of different frequencies, strictly speaking, are not an indicator of the role of this structure in the formation of the spectrum of the calling song during stridulation of a living insect. Bailey also used isolated tegmina, but imitated their movements during singing using a special device.
The authors operate with the values of the resonant frequencies of pinnae cavities, obtained either from calculations that are used to determine resonances in Helmholtz resonators, or experimentally when studying enlarged 3D models of the proximal part of foreleg of katydids. They do not take into account (or do not mention) the role of the intrinsic resonances of the pinnae cavitiy walls (tympanal membranes and pinnae) and their possibility of absorption of sound vibrations. At a wavelength comparable to the size of the tympanal slits, the sound enters the cavity, where the own resonances of the cavity walls are observed (if they exist), as well as reflection, re-reflection of sound and its absorption by the walls. It should be remembered that the absorption capacity depends not only on the nature of the material of the cavity walls, but also on the sound frequency. The higher the carrier frequency of the sound wave, the higher it is. Thus, it is not entirely correct to extrapolate the data obtained from 3D models to the natural auditory organs of katydids, because the materials of the model and the tissues of living insects have different physical characteristics, including acoustic ones. The role of pinnae in the perception of sound stimuli can be clarified with much greater reliability by recording the whole-tympanal nerve activity in an intact insect and after removing the pinnae.
It is unclear whether the given resonant frequencies of the cavity under the ears are calculated, or obtained experimentally on 3D models.

Validity of the findings

No comments apart from the above

Additional comments

Detailed comments
L 48-49 –please insert e.g.
Currently, this is a well-known fact and there are a huge number of articles devoted to this issue.
Besides, sometimes katydids have two or even three sound organs, which they use to produce different types of sounds. The following stridulation devices are currently described: sternocoxal in Phyllophorinae (Dumortier, 1963; Lloyd, 1976, Korsunovskaya et al. 2021); abdominal-alar in Pantecphylus cerambycinus (Pseudophyllinae) and Mygalopsis marki (Conocephalinae) (Heller, 1996; Bailey, Sandow, 1983); labral (Mecopodinae) (Lloyd, Gurney, 1975).
L 65-66 - please add the Lewis article (1974a ), who was the first to point out the importance of the acoustic trachea for sound perception in bush-crickets.
L 72 - protection of the pinnae – may be tympanal membranes?
L 77-80 - Please rewrite the phrase
L 117 add n. sp.
L 279 - it is advisable to give drawing of titillators
L 283 Pronotum buff – Is it disc of pronotum?
L 304 - It is not clear what signal is being described. It is not indicated what quantitative parameters are given. The ~ sign is not allowed when indicating the mean and SD (or standard error?) 37.4 ± 7 ms. Please indicate this in METHODS
L 310, 376 r.e. change to re
L 374-375 “Tracking of the wing motion of S. sodalis reveals that sound is only produced during wing closure (Fig. 11)”. - MATERIAL AND METHODS does not contain a description of the method used to track the movements of the tegmina during singing

L 408 styli in katydids are paired structures, you can distinguish the place from which the stylus broke off
L 425 unnecessary parentheses
L 456 - microseconds >> milliseconds?
L 458 “They also manage to conserve purity in their calls” - it is not clear what the authors mean when they write about the purity of the call
L 464, 474 - sinusoidal pulses - these are not sinusoidal pulses. Sinusoidal - see in Montealegre-Z & Morris, 1999 (e.g. Fig. 32B)
L 468-469 Author’s reasoning about the similarity of the spectra is entirely based on the structure of the stridulatory files. However, the spectra can be influenced to the same extent by the resonant properties of other parts of the tegmina.
L 477-479 “carrier frequency peak of 13.4 kHz” and “slightly higher carrier frequency peak of 16.3 kHz”. - In this case, the comparison of amplitudes is incorrect, because the amplitude of the spectral maxima depend on the equipment used and on the distance of the microphone to the singing insect. In addition, on the Y-axis of Fig. 10C authors did not indicate units of measurement.
L 518 ~ 120 – 60 kHz: vice versa?
L 543-544 Geographical distribution sets – please change sets to areas or exclude it
557-559 – “Coupled with similarities in acoustic data, we propose that be placed as sister genus of Satizabalus be placed as sister genus of Trichotettix within the Cocconotini tribe”.
It seems to me that it is better to exclude this sentence, since the similarities and differences of the two taxa can only be judged through a more detailed and in-depth comparative analysis
L 609 and onwards. Biology in captivity - Transfer to METHODS or RESULTS

L 667 change tier und mensch to Tier und Mensch
L672 change terminal to tegminal
L 682 Kaiser-lich-Konigliche – please check this
L 773 delete D,

FIGURE LEGENDS

Figs 3, 8, 12 Change:
Anatomical features to Morphological characters;
Right leg tympanum. – to proximal part of foreleg;
Stylus to styli; Full female specimen – to habitus of female.
Figs 6C, 10C – add units of measurement on the Y-axis
High resolution… unfortunate term. Possible variant: oscillogram with a higher speed
Fig. 17 – please provide a full explanation of the abbreviations ATM and RTM
Fig. 18. Change c.a. to ca (ca – from latin circa).

---

## Round 0.2 · Minor Revisions

Please, consider the suggestions of reviewer 3.

·

Basic reporting

See previous review. Problems removed

Experimental design

See previous review. Problems removed

Validity of the findings

See previous review. Problems removed

Additional comments

See previous review. Problems removed

·

Basic reporting

The revised version seems to be okay (no time to read everything carefully). See additional comments.

Experimental design

n/a

Validity of the findings

n/a

Additional comments

suggested corrections according to line numbers

282 n. gen. (space)

288 Satizabalus jorgevargasi n. sp. (as with the other new species)

289 Satizabalus huaca n. sp. (space)

292 (present … grooves

295 In the key to Eucocconotini genera … characters (

342 (without mentioning authors it would be S. jorgevargasi Holmes, Woodrow, Sarria-S., Celiker & Monetalegre-Z., 2024)

342 n. sp. (space)

496 (without mentioning authors it would be S. huaca Holmes, Woodrow, Sarria-S., Celiker & Monetalegre-Z., 2024)
496 n. sp. (space)

677, 681 ter Hofstede

943 http://orthoptera.archive.speciesfile.org (this is now the URL of the old OSF accessed in April 2023)

1044, 1047, 1051 ter Hofstede (in the reference sections of her own papers she spells the prefix with lower case, also as first author)

Reviewer 3 ·

Basic reporting

The necessary corrections, additions and clarifications have been made to the article.

Experimental design

Minor comments regarding the title of the article. The author, at the request of the reviewer, replaced the word “bioacoustics” with acoustic communication. However, acoustic communication involves both sound signals and hearing. Therefore, it is advisable to either exclude the word “hearing” from the title, or replace “communica tion” with sound production (or acoustic signals or signaling).
L 163-164 Please correct tegmen were… with tegmen was…
The wing fixation procedure is important for interpreting the results obtained. It should be described in more detail, in particular, indicate the area of the elytra covered with wax.

Validity of the findings

As mentioned in Basic reporting

---

## Round 0.3 · accepted · Accept

The authors provided all necessary changes. The paper can be accepted.